PREREGISTERED RESEARCH ARTICLE

# The effect of apathy and compulsivity on planning and stopping in sequential decision-making

Jacqueline Scholl [1,2,3,4] *, Hailey A. Trier [3], Matthew F. S. Rushworth [3,5�‡], Nils Kolling [4,6☱‡]

**1** Lyon Neuroscience Research Center, INSERM U1028, CNRS UMR5292, PSYR2 Team, University Lyon 1, Lyon, France, **2** Centre Hospitalier Le Vinatier, Pôle EST, Bron, France, **3** Wellcome Integrative Neuroimaging (WIN), Department of Experimental Psychology, University of Oxford, Oxford, United Kingdom, **4** Oxford Centre of Human Brain Activity, Wellcome Integrative Neuroimaging (WIN), Department of Psychiatry, University of Oxford, Oxford, United Kingdom, **5** Centre for Functional MRI of the Brain (FMRIB), Wellcome Integrative Neuroimaging (WIN), Nuffield Department of Clinical Neurosciences, University of Oxford, Oxford, United Kingdom, **6** Univ Lyon, Université Lyon 1, Inserm, Stem Cell and Brain Research Institute U1208, Bron, France

☱ These authors contributed equally to this work.
‡ These authors are co-senior authors on this work.
* Jacquie.scholl@gmail.com

**Note:** As this is a Preregistered Research Article, the study design and methods were peer-reviewed before data collection. The time to acceptance includes the experimental time taken to perform the study. Learn more about Preregistered Research Articles.

## Abstract

Real-life decision-making often comprises sequences of successive decisions about whether to take opportunities as they are encountered or keep searching for better ones instead. We investigated individual differences related to such sequential decision-making and link them especially to apathy and compulsivity in a large online sample (discovery sample: $n = 449$ and confirmation sample: $n = 756$). Our cognitive model revealed distinct changes in the way participants evaluated their environments and planned their own future behaviour. Apathy was linked to decision inertia, i.e., automatically persisting with a sequence of searches for longer than appropriate given the value of searching. Thus, despite being less motivated, they did not avoid the effort associated with longer searches. In contrast, compulsivity was linked to self-reported insensitivity to the cost of continuing with a sequence of searches. The objective measures of behavioural cost insensitivity were clearly linked to compulsivity only in the discovery sample. While the confirmation sample showed a similar effect, it did not reach significance. Nevertheless, in both samples, participants reported awareness of such bias (experienced as "overchasing"). In addition, this awareness made them report preemptively avoiding situations related to the bias. However, we found no evidence of them actually preempting more in the task, which might mean a misalignment of their metacognitive beliefs or that our behavioural measures were incomplete. In summary, individual variation in distinct, fundamental aspects of sequential decision-making can be linked to variation in 2 measures of behavioural traits associated with psychological illness in the normal population.

reproduction in any medium, provided the original
author and source are credited.

**Data Availability Statement:** All anonymized data
and analyses script are freely available on a
repository (https://osf.io/dfg2u/). The data are also
available in the supplementary data files (S1–S16
Data).

**Funding:** This work was funded by the Medical
Research Council (MR/N014448/1, Skills
Development Fellowship,JS; G0902373, grant
MFSR, https://mrc.ukri.org/), the Biotechnology
and Biological Sciences Research Council (BB/
R01803/1, BB/V004999/1, AFL Fellowship NK;
Discovery Fellowship JS, https://bbsrc.ukri.org/),
the Clarendon Fund (studentship, HAT https://
www.ox.ac.uk/clarendon) and the Wellcome Trust
(WT100973AIA and 203139/Z/16/Z, https://
wellcome.org/ MFSR). The funders had no role in
study design, data collection and analysis, decision
to publish, or preparation of the manuscript.

**Competing interests:** The authors have declared
that no competing interest exist.

**Abbreviations:** AMI, Apathy Motivation Index; BDI,
Beck Depression Inventory; CFI, comparative fit
index; GCSE, General Certificate of Secondary
Education; OCD, obsessive-compulsive disorder;
OCI-R, Obsessive-Compulsive Inventory; RMSEA,
root mean square error of approximation; RT,
reaction time; SRMR, standardised root mean
square residual.

## Introduction

Real-life decision-making behaviour often involves planning a sequence of decisions and then carrying them out while monitoring their appropriateness. Such sequential decision-making patterns, rather than single event decisions, often characterise the decision-making problems that have to be solved by foraging animals and, equally, by humans. While sequential decisions are ubiquitous in nature, very little attention has been paid to cognitive models of decision-making that take sequential context and planning into account. This is an important issue as many decision problems, and the cognitive mechanisms needed to deal with them, are unique to sequential decisions [1–3]. We illustrate an example of a sequential job search in Fig 1. Psychologically and neurally, the processes involved here are distinct from more commonly studied single event decisions [4–10]. It has been argued that this kind of paradigm is more ecological and captures aspects of decisions in real life particularly well [11–13]. If this is the case, then it is possible that such tasks will probe individual differences in decision-making that are related to meaningful individual differences in psychological traits that, in the extreme, are related to psychological illness.

To examine this possibility, here, we have paired a previously established "sequential search task" [14] with measures of apathy and compulsivity—traits that are common features of psychological illness collected in a large online sample in a transdiagnostic approach [15]. In addition to the behavioural measures intrinsic to the sequential task, we have also collected a set of introspective and self-report measures relating to the task. Crucially, we do not just examine raw behavioural scores, but, instead, we use a computational model to obtain precise and quantitative estimates of the cognitive processes that led to the participants' behaviour in the task. They capture firstly how well participants plan a sequence of searches and secondly how they then carry out the sequence of searches. We can obtain measures of both rational and irrational behaviour. We can measure to what extent participants can flexibly adjust their pre-planned behaviours "on the fly." We can also measure how participants fail to adjust through 2 biases (Fig 1F): First, participants can be insensitive to repeatedly incurred costs of decisions as they progress through a sequence. Second, participants can have decision inertia, i.e., inflexibility and repetitiveness. This is distinct from action inertia, which would lead to increased reaction times (RTs) but no change in which decisions are taken. Fig 1F illustrates both of these biases. Sensitivity to costs is the degree to which searching is made less likely by increasing costs. Decision inertia is the likelihood of deciding to search yet again because you have searched already; when there is decision inertia, then the higher the previous numbers of searches, then the more likely you should be to search yet again. Formally, we can use our computational models to get quantitative measures of each of these biases, while controlling for all other factors that might affect choices. Finally, we can also measure how participants' knowledge of their own biases enables them to preemptively avoid situations that are more likely to trigger a bias. We measure this here by asking participants to self-report whether they preemptively avoided these situations.

Making categorical comparisons between patient groups and controls is an important line of research that has delivered major insights into changes in cognition and behaviour in psychological illness. An alternative approach is to use a transdiagnostic dimensionality-based approach. We chose this approach here due to its ability to extract multiple distinct dimensions within large population data and across different questionnaires and traditional diagnostic boundaries [16,17]. Specifically, we collected questionnaires with measures covering a wide range of psychiatric symptoms and then extracted symptom dimensions cutting across traditional disorder boundaries. This is particularly relevant for 3 reasons. First, many disorders, such as obsessive-compulsive disorder (OCD), have comorbidity rates as high as 90% [18].

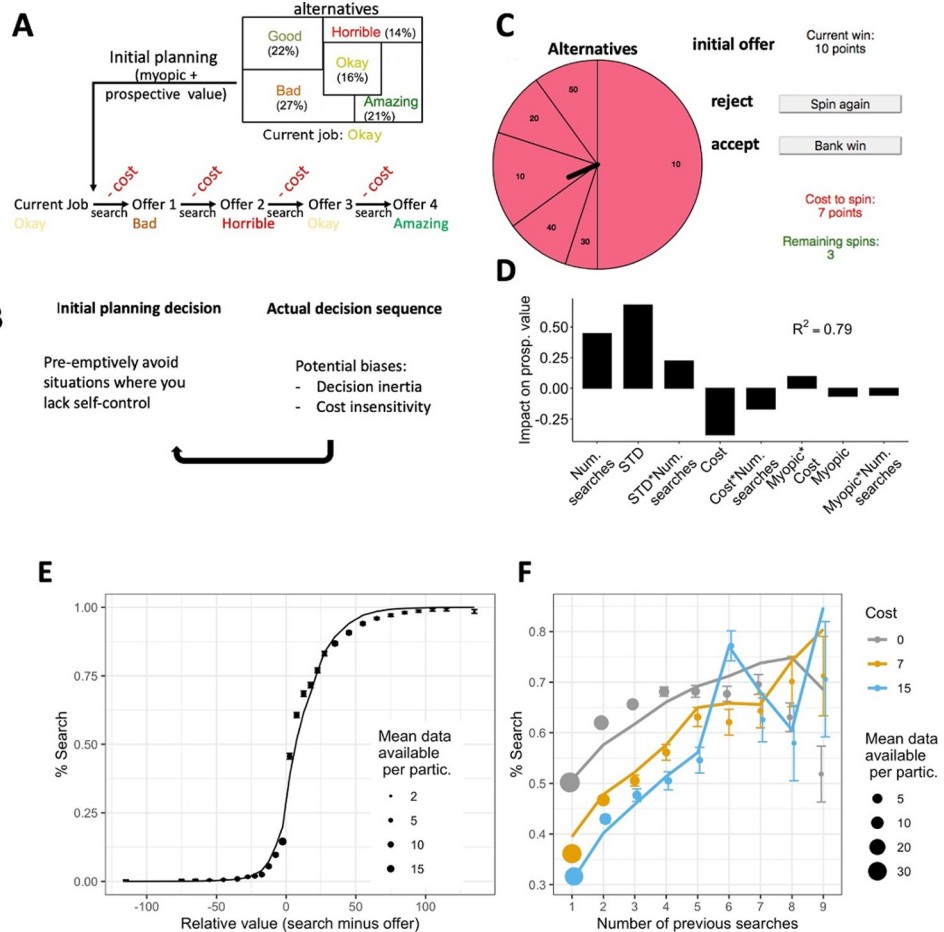

Fig 1. Sequential search paradigm. (Ai) Deciding whether to leave a job can be seen as a sequential search problem in which one repeatedly encounters alternative job offers, one at a time, that one can then either accept or instead reject in order to search for another alternative from the job market. In short, the decision-maker needs to weigh up the value of the current offer with the value of the job market. The job market's value has 2 important components. First, it consists of the average ("myopic value") value of the alternatives in the job market [14]. Second, it consists of its "prospective value," i.e., the value that stems from the possibility to make sequences of searches within this pool of alternatives [14]. Taking prospective value into account means that the value of the alternatives is higher than might initial be thought on the basis of the average, myopic value. The degree to which this is the case depends on how many opportunities one has to explore the alternatives and the cost of doing this (see S1 Text "Decision tree model to derive prospective value" for more details). In the illustrated sequence, searches were continued until a sufficiently good alternative was encountered (one has found one's "dream job"). Instead, if the opportunities for exploring are limited (i.e., only a limited number of searches are available), searches might have to stop when the available opportunities to search are depleted, before a "dream job" was found. (B) Sequential decisions can be broken down into 2 stages. The first is the initial consideration of the environment and its potential outcomes. Second, the agent goes through the actual sequence of searches. Here, biases of insensitivity to search cost or decision inertia can emerge. If agents are aware of these biases, they can be preemptively avoided by not engaging with sequences in the first place especially if the potential sequence is of the type in which one may be more prone to the biases or in which the biases may be more costly. (C) Illustration of the actual task screen participants saw. The wheel of fortune shows all possible alternatives and the sizes of the areas represent their probabilities of being drawn. Participants can accept the initial offer or any later offer by clicking "bank win" or search for an alternative by deciding to "spin again" that will reduce the number of "remaining spins" by one and cost as many points as show as "cost to spin." (D) Prospective value is derived from a decision tree model (see S1 Text). This model captures the way in which prospective value is higher in environments with many available searches (num. searches), higher variability in how good and likely the offers are (STD), particularly if many searches are available (STD*Num. searches) and low cost, again particularly if many searches are available. Results are regression weights relating prospective value to different features of the environment (see Methods). The proportion of variance of prospective value explained by the regressors ($R^2$) was 79%. (E) Participants' behaviour (confirmation sample—same results were found in the discovery sample) and model predictions for decisions after the initial planning stage: The higher the relative value favouring searching versus accepting the offer, the more likely participants were to search (dots with error bars). The model's behaviour matches participants' very

closely (line). **(F)** Illustration of the main behavioural measures in the confirmation sample used to relate clinical factors to. In general, the more often participants (dots with error bars; continuous lines show model predictions) have already searched on a given trial, the more likely they were to search again, rather than accept an offer ("decision inertia"). Note that in contrast to the regression models (S1 and S2 Figs), this split does not correct for the fact that other factors such as search value are partially correlated with the number of previous searches, but it does provide a simple way to visualise the impact of task features on participant behaviour. The graph also shows the higher the cost (grey to blue), the less likely participants were to search. Error bars show the standard error of the mean, size of the dots show the mean number of data points per participant, and lines show the model predictions. Data in panel E are available in S1 Data and panel F in S8 Data. STD, standard deviation.

Secondly, it is well known that some dimensions are shared across many disorders [19]. Third, recent work [20] suggests that dimensions cutting across disorders relate more strongly to behavioural deficits than diagnostic categories. Here, we are particularly interested in the dimensions of apathy and compulsivity and want to study them while controlling for other confounding factors. In general terms, the hypotheses that we want to test are as follows (for exact statistical tests, see Methods).

First, we hypothesise apathy will be associated with decision inertia. Apathy exists on a spectrum and in its more severe form is a symptom present in many disorders typically regarded as distinct. It is prominent in depression, in the negative symptoms of schizophrenia, after specific brain injuries [19], and in burnout [21]. It is sometimes described as avolition because of the problems patients can have with initiating new behaviours [22] or generating new options as candidates for decision-making [23]. If the option of stopping a sequence of decisions, such as job search, does not even occur to you then you might exhibit decision inertia, continuing to search and exerting more and more effort (i.e., "overcommitting" [24]). In this way, apathy could counterintuitively lead to the exertion of more unnecessary effort, which, in turn, might increase feelings of apathy. Therefore, we predict that apathy will be linked to an increased decision inertia (**hypothesis 1**), i.e., a bias to search again and again for a new offer without considering that the appropriate strategy now would be to stop the search. One could also call this a commitment or closure inertia. Note that the hypothesis is very distinct to the more simplistic idea that apathy just occurs at the action level as a failure to act quickly, which could express itself as slower response times. Instead, we are proposing it occurs at the decision level and therefore that it may even lead to the commission of more actions.

Second, we wanted to investigate effects of the compulsivity trait on sequential search behaviour. Like apathy, compulsivity exists on a spectrum and is increased in several disorders such as OCD [25], eating disorders [26], and addiction [27]. Particularly for OCD, compulsivity is seen as central: OCD is characterised by obsessions and compulsions (e.g., obsessions about cleanliness and compulsive handwashing). For an observer, such behaviour appears to lack utility, but patients with OCD carry out the compulsive behaviours despite their immense time costs [28]. One important dimension determining utility is the cost of a behaviour. Insensitivity to the costs of the compulsive behaviour could then lead to impairments in arbitration between more cognitively demanding model-based (or "goal directed") and model-free (or "habitual") modes of behavioural control. This leads us to our second hypothesis: OCD-related compulsivity (measured here using questions from OCD and schizotypy inventories; "compulsivity" for short) will be linked to cost insensitivity when carrying out a sequence of searches and that participants will be aware of problems linked to this bias ("overchasing," **hypothesis 2**). Note that this approach is consistent with recent evidence that obsessions may not be the primary features of OCD that cause the occurrence of obsessions [29–31].

Participants' awareness of their own bias directly leads to our next question. If compulsivity is linked to an awareness of the risk of "overchasing," do participants do anything to try to

compensate for this? Interestingly, we know that OCD is an "ego-dystonic" disorder, i.e., despite awareness, a patient is not able to inhibit their obsessions or compulsions. However, sometimes, a patient may be able to avoid situations that trigger the obsession–compulsion complex altogether; for example, family members may accommodate OCD sufferers by doing the washing instead of them to avoid triggering the OCD [32]. Here, we wanted to examine the relationship between OCD symptoms and the presence of preemptive decision-making strategies that occur at the beginnings of search sequences and which prevent the person from entering situations later in the sequence in which they would be likely to perform poorly because of a bias in decision-making. We hypothesise that OCD symptoms are associated with self-reported avoidance of situations in which biases will manifest (**hypothesis 3**).

For data collection and analysis, we employed a "split sample" approach [33,34]. We first collected a discovery sample that we used to refine and develop hypotheses in a data-driven way. We then used a confirmation sample to test the preregistered hypotheses in a statistically rigorous way. This approach thus combines the benefits of preregistration—reduced risk of false positives due to analytical freedoms—with those of exploratory science—the analytical approach being best adapted to the specific data.

## Results

A total of 830 participants completed all parts of this preregistered study (link to OSF https://osf.io/dfg2u). Exclusions: 33 participants (4%) due behavioural responses suggesting lack of task understanding; 4 participants (0.5%) due to not completing all questionnaires or failing the questionnaire attention checks and 112 participants (13%) from the behavioural analyses of initial searches; and 44 participants (5%) from later searches due to too little variation in their responses to allow analysis (see Methods). This resulted in 756 participants included for preregistered analyses linking transdiagnostic dimensions and behaviour and 795 participants included in analyses of only the questionnaires (demographics in Table 1).

Power analyses of the discovery sample ($n = 449$ for analyses linking transdiagnostic dimensions and behaviour; see link/reference for detailed results of the discovery sample data submitted for preregistration) had suggested a sample size of 750 participants to achieve 95% power for all preregistered analyses. Throughout, we report results from the confirmation sample. Analyses that deviated from the preregistration are highlighted as "exploratory." Results of the discovery sample can be seen the preregistration (https://osf.io/dfg2u). The questionnaires from the discovery sample data have since been published as part of a meta-analysis on the relationship between apathy and compulsivity I large population samples [35].

On each of 150 trials of the task, participants chose to either "bank" an "initial offer" or carry out the decision sequence and search among the alternatives by "spinning a wheel of fortune" (Fig 1C). Their aim was to earn as many points (i.e., money) as possible. If they banked the offer (i.e., collected the number of points of the offer), they moved on to the next trial. If instead they decided to "spin," they had to pay a small cost and could then spin the wheel to get a new offer. In each round, there was a maximum number of times participants could spin the wheel again if they were not satisfied with their draws; each time they had to pay the same cost. Participants were shown the available offers on the wheel, how likely they were (size of area on wheel), how much it costs to spin the wheel, and how often they could spin the wheel on a trial (also see Methods for more details on properties of trials). After the task, participants were asked to introspect about their performance in the task (see Methods). We split the behavioural data into initial and later searches to be able to test effects of prospection at the first decision separately from sequential decision effects such as perseverance.

**Table 1. Demographics and experiment duration for participants of the confirmation sample that we included or excluded from all analyses based on questionnaire and task performance criteria (see Methods).**

|  | Excl. not enough variation in later decision responses N = 39 | Excl. questionnaires N = 2 | Excl. task understanding N = 33 | Included N = 756 | p-Value |
|---|---|---|---|---|---|
| Age | 29.4 (5.99) | 26.5 (7.78) | 29.9 (6.14) | 28.2 (6.32) | 0.298 |
| Gender |  |  |  |  | 0.372 |
| Female | 24 (61.5%) | 0 (0.00%) | 18 (54.5%) | 385 (50.9%) |  |
| Male | 14 (35.9%) | 2 (100%) | 15 (45.5%) | 361 (47.8%) |  |
| Other | 1 (2.56%) | 0 (0.00%) | 0 (0.00%) | 10 (1.32%) |  |
| Education |  |  |  |  |  |
| GCSE | 2 (5.13%) | 0 (0.00%) | 2 (6.06%) | 39 (5.16%) |  |
| A-levels | 17 (43.6%) | 1 (50.0%) | 7 (21.2%) | 220 (29.1%) |  |
| Bachelor | 11 (28.2%) | 1 (50.0%) | 11 (33.3%) | 319 (42.2%) |  |
| Master's | 6 (15.4%) | 0 (0.00%) | 11 (33.3%) | 157 (20.8%) |  |
| Doctorate | 3 (7.69%) | 0 (0.00%) | 2 (6.06%) | 21 (2.78%) |  |
| English fluency |  |  |  |  | 0.016 |
| Basic | 1 (2.56%) | 0 (0.00%) | 2 (6.25%) | 4 (0.53%) |  |
| Moderate | 1 (2.56%) | 0 (0.00%) | 5 (15.6%) | 62 (8.20%) |  |
| Native | 37 (94.9%) | 2 (100%) | 25 (78.1%) | 690 (91.3%) |  |
| Medication status |  |  |  |  | 0.042 |
| No | 37 (94.9%) | 1 (50.0%) | 33 (100%) | 682 (90.2%) |  |
| Yes | 2 (5.13%) | 1 (50.0%) | 0 (0.00%) | 74 (9.79%) |  |
| Experiment duration (min) | 65.5 (29.5) | 65.2 (9.43) | 83.2 (29.0) | 64.4 (18.4) | <0.001 |
| Task duration (min) | 34.9 (17.8) | 38.7 (1.88) | 44.3 (17.5) | 36.0 (11.4) | 0.002 |
| Questionnaires duration (min) | 18.9 (21.3) | 15.0 (2.11) | 20.5 (13.9) | 16.4 (7.62) | 0.026 |
| Training duration (min) | 6.58 (4.50) | 8.66 (5.54) | 11.3 (9.07) | 6.70 (6.52) | 0.001 |

For a detailed list of medications, see S9 Table. Data shown in this table are available in S4 Data.

GCSE, General Certificate of Secondary Education.

## Validation of sequential decision task

Detailed validation of the paradigm and description of all model parameters can be found in S1 Text and S1 and S2 Figs. The model captured participants' behaviour well (Fig 1E and 1F). In the analyses, we related different factors that should rationally or irrationally drive behaviour (e.g., value of the offer and cost of searching). We examined both participants' decisions either to search or accept an offer and their RTs when they made these decisions. In short, we found first that participants planned ahead (S1A Fig, parameter "Prospective"). Second, when executing a sequence, participants were sensitive to the value and cost of searching (Fig 2A, parameters "Myopic," "ProspVS1," and "Cost") and showed flexible adaptation of their strategy throughout the sequence (Fig 2A, parameter "ProspVS1-Adapted"), but also irrational biases that led them to search too many times. In particular they showed decision inertia, i.e., a "stuck in the rut" bias: The more participants had already searched, the more likely they were to search again, beyond what they should have done given the value of the alternatives (Fig 2A, parameter "# Prev Searches"). Participants also self-reported awareness of this bias (Fig 2B). Another bias was cost insensitivity during repeated searches, i.e., some participants did not take the repeatedly encountered costs sufficiently into account (Fig 2A, parameter "Cost"). Relatedly, participants reported "overchasing" of alternatives, i.e., searching too often because one of the alternatives appeared particularly appealing (Fig 2C). Third, we found that

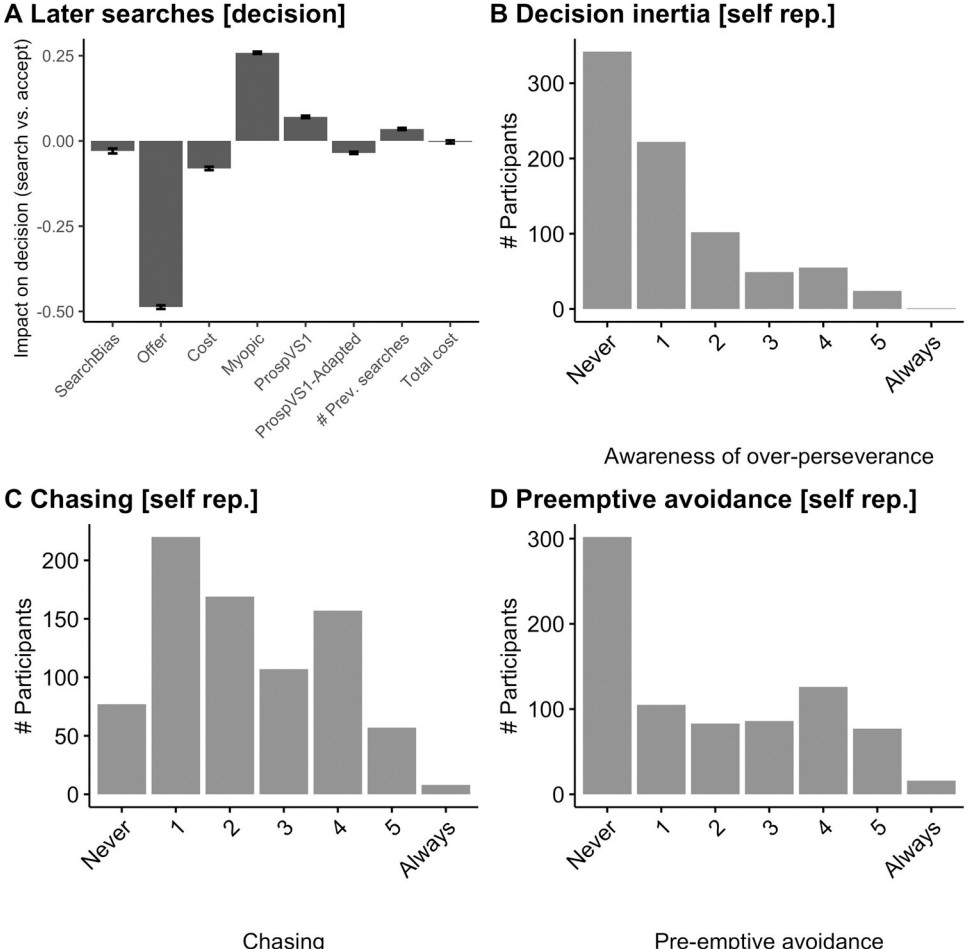

**Fig 2. Group-level results (confirmation sample). (A)** After initiating a search sequence, participants made a series of decisions of either banking an offer (and ending the trial) or searching again. We can assess, using our computational model (methods section "Analysis—Decisions"), whether participants behave mostly rationally and also whether they have decision inertia, i.e., show a "stuck in the rut" bias or a cost insensitivity bias. As in the discovery sample and as they should rationally, we found participants were more likely to go on searching if the myopic value was high (mean = 0.26, 95% CI [0.26 to 0.26]), if the initial prospective value ("ProspVS1"; i.e., the prospective value at the first search on each trial) was high (mean = 0.07, 95% CI [0.07 to 0.07]) or if the cost of searching was low (mean = −0.08, 95% CI [−0.08 to −0.08]) or if the drawn offers were low (mean = −0.49, 95% CI [−0.49 to −0.48]). Participants also adjust their estimate of prospective value ("ProspVS1-Adapted") as it decreased over time as fewer and fewer available searches remained in a trial (mean = −0.03, 95% CI [−0.04 to −0.03]). We also found evidence for decision inertia ("stuck in a rut" bias, "# Prev. searches"): The more participants had already searched, the more likely they were to search again (mean = 0.04, 95% CI [0.03 to 0.04]). We tested participants sunk cost fallacy, i.e., whether they were more likely to continue searching the more costs they had already paid on a trial. However, this was not the case ("TotalCost" is not positive, (mean = −0.00, 95% CI [−0.00 to −0.00]). **(B)** Despite the strong behavioural effects of decision inertia, about half of the participants reported having no such bias (self-report Q 1). **(C)** Contrary to this, about 80% of people reported having been biased towards overchasing a rewarding option at least somewhat (self-report Q2). **(D)** About half of participants reported preemptively avoiding trials when they were worried they might spin the wheel too many times (self-report Q3). Error bars show Bayesian 95% credible intervals (2 tailed), significance is shown by credible intervals not including zero. Data in panel A are available in S9 Data (raw data in S1 Data) and panels B to D in S2 Data.

participants preemptively avoided trials on which a bias to search too many times would be particularly costly (S1A Fig, parameter "AvgProspVChange"). They self-reported awareness of this preemptive strategy (Fig 2D). For robustness of our modelling approach, see parameter recovery in Table A in S1 Text and S5 Fig (same data for real participants in Tables B and C in

S1 Text). The results of the confirmation sample replicate almost all results from the discovery sample: For models of participants' decisions (Fig 2A as well as S1A and S2A Figs), 13 out of 14 significant results replicated, i.e., the same factors that drove choices in the discovery sample were significant in the confirmation sample with the exception that in the discovery sample, accumulated cost during later searches on each trial reduced the probability of searching again, while it did not affect behaviour in the confirmation sample. For a description of those behavioural results, see the figure legend of Fig 2 as well as S1 and S2 Figs. For models of RT (S1B and S2B Figs), 10 out of 11 significant results replicated (exception: in the discovery sample, initial prospective value speeded up responses during later searches, while it did not impact behaviour in the confirmation sample).

## Clinical profile and dimensions

Participants were not specifically selected to have a minimum level of psychiatric symptoms (Table 1). Nevertheless, a sizable proportion of self-reported symptoms were in a clinically significant range using standard clinical cutoffs (Fig 3A, Table D in S1 Text), similar to previous findings in online samples [36,37]. For example, 30% of participants scored as having moderate or severe depression, and 43% of participants scored as having significant OCD symptoms.

We derived transdiagnostic dimensions cutting across 8 questionnaires (23 subscales, Fig 3B). Specifically, in the discovery sample (S3 Fig), we performed an exploratory factor analysis on the standard subscale summary scores extracted from each questionnaire. This resulted in 4 factors.

Our 4 factors could be labelled "compulsivity," "apathy," "depression/anxiety," and "social anxiety." The compulsivity factor mainly comprised subscales from the Obsessive-Compulsive Inventory (OCI-R) questionnaire (with a notable exception of the "obsessing" subscale) and the subscale measuring "unusual experiences" (e.g., hearing voices or telepathy) from the schizotypy questionnaire. The apathy factor included 2 of the 3 subscales from the Apathy Motivation Index (AMI): social and emotional apathy, but not the third subscale, behavioural apathy; it also included the anhedonia subscale of the schizotypy questionnaire and the "lack of importance of emotions/externally oriented thinking" from the Toronto Alexithymia questionnaire (including items such as "being in touch with emotions is essential").

As the factor analysis used oblique rotation, the factor scores between dimensions were somewhat correlated (Table 2, highest correlation $r = 0.65$ between social anxiety and depression/anxiety, all others $r < 0.38$).

## Validation of factor analysis

We validated the factor analysis in the confirmation sample using confirmatory and exploratory factor analysis.

First, we ran a confirmatory factor analysis. Our comparative fit index (CFI) was 0.815, which indicates moderate, but not great fit [68]. However, our standardised root mean square residual (SRMR) and root mean square error of approximation (RMSEA) was relatively high at 0.097 and 0.112, respectively, indicating a relatively poor fit. This could be because of differences in how the exploratory and confirmatory factor analysis extracted dimensions. For example, the confirmatory factor analysis required a categorial assignment of each questionnaire subscale to a specific factor, while our questionnaire subscales could sometimes load onto more than one dimension (Fig 3C, S3C Fig). Therefore, we checked reliability of our factor solution further using exploratory factor analysis.

Specifically, we ran an exploratory factor analysis on the confirmation sample, resulting in 4 factors, as in the discovery sample. Visual inspection of the loadings of subscales to factors in

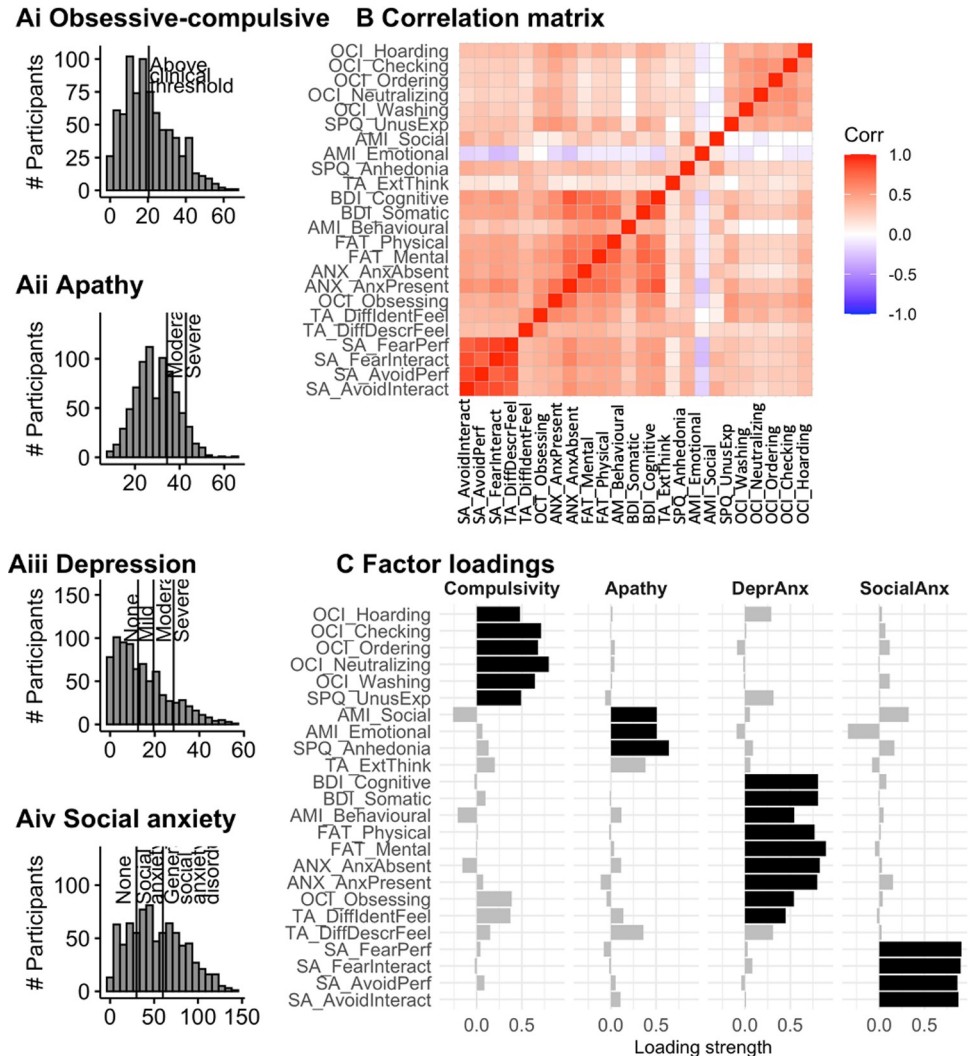

**Fig 3. Psychiatric symptom questionnaires in the confirmation sample. (A)** Histograms of the distribution of total questionnaire scores for the OCI-R (Ai), the AMI (Aii), the BDI (Aiii), and the Liebowitz social anxiety scale (Aiv). On each histogram, cutoffs based on previously published normative data are highlighted. **(B)** Correlation matrix (Pearson's r) for all subscales included in the factor analysis. Squares highlight the factors subscales were assigned to in the factor analysis. **(C)** Factor loadings (i.e., the contribution of each subscale to each factor) of the factor analysis of the subscales. Highlighted in black are loadings above 0.4, for ease of visualisation. Data in panel A are available in S5 Data, panel B in S3 Data, and panel C in S16 Data. AMI, Apathy Motivation Index; BDI, Beck Depression Inventory; OCI-R, Obsessive-Compulsive Inventory.

the confirmation (Fig 3C) and discovery sample (S3 Fig) showed that both were virtually identical. To quantify this, we computed the factor scores for each person (e.g., a person's "compulsivity score") in the confirmation sample for each factor based either on factor loadings (i.e., how much each subscales contributes to a factor) from the exploratory factor analysis of the discovery sample or of the confirmation sample. Confirming the impression from the similarity of the factor weights in both analysis, these were extremely highly correlated (r between 0.96 and 1; Table 2).

Therefore, we next proceeded with our preregistered analysis strategy of applying the factor weights from the discovery sample to the confirmation sample to obtain factor scores for each person and factor. We then related these transdiagnostic dimensions to behavioural measures.

**Table 2. Correlation between factor scores (\*\*\*$p < 0.001$) for the confirmation sample based on factor loadings from either exploratory factor analysis done on the confirmation sample ("[C]") or based exploratory factor analysis from the discovery sample ("[D]").**

| | DA[C] | SA[C] | C[C] | A[C] | DA[D] | SA[D] | C[D] | A[D] |
|---|---|---|---|---|---|---|---|---|
| *DeprAnx (DA)[C]* | | 0.60\*\*\* | 0.36\*\*\* | 0.23\*\*\* | 1.00\*\*\* | 0.62\*\*\* | 0.38\*\*\* | 0.14\*\*\* |
| *SocialAnx (SA)[C]* | 0.60\*\*\* | | 0.30\*\*\* | 0.25\*\*\* | 0.63\*\*\* | 0.99\*\*\* | 0.34\*\*\* | 0.16\*\*\* |
| *Compulsivity (C)[C]* | 0.36\*\*\* | 0.30\*\*\* | | 0.03 | 0.38\*\*\* | 0.32\*\*\* | 0.99\*\*\* | -0.06 |
| *Apathy (A)[C]* | 0.23\*\*\* | 0.25\*\*\* | 0.03 | | 0.27\*\*\* | 0.26\*\*\* | 0.05 | 0.96\*\*\* |
| *DeprAnx[D]* | 1.00\*\*\* | 0.63\*\*\* | 0.38\*\*\* | 0.27\*\*\* | | 0.65\*\*\* | 0.40\*\*\* | 0.17\*\*\* |
| *SocialAnx[D]* | 0.62\*\*\* | 0.99\*\*\* | 0.32\*\*\* | 0.26\*\*\* | 0.65\*\*\* | | 0.36\*\*\* | 0.17\*\*\* |
| *Compulsivity[D]* | 0.38\*\*\* | 0.34\*\*\* | 0.99\*\*\* | 0.05 | 0.40\*\*\* | 0.36\*\*\* | | -0.05 |
| *Apathy[D]* | 0.14\*\*\* | 0.16\*\*\* | -0.06 | 0.96\*\*\* | 0.17\*\*\* | 0.17\*\*\* | -0.05 | |

*Computed correlation used Pearson method with listwise deletion*

Both solutions were virtually identical (correlations between e.g., Apathy[C] and Apathy[D]; all $r > 0.96$). Neither compulsivity nor apathy were correlated to any of the other factors at more than $r > 0.38$ (i.e., no shared variance $r^2 > 14.4\%$). Nonetheless, in all analyses, we controlled for all transdiagnostic dimensions. Data shown in this table are available in S16 Data.

## Testing of preregistered hypotheses in the confirmation sample

Next, we tested the relationship between the transdiagnostic dimensions of apathy and compulsivity with the behavioural and self-report measures described above. We simultaneously controlled for demographic measures (age, gender, and education), as well as psychiatric factors of no interest (depression/anxiety and social anxiety) and general measures of behavioural stochasticity in the sequential decision-making task (e.g., inverse temperature, as a measure of choice noisiness). For RT analyses, we controlled for average RT and RT variability. We only report results of preregistered analyses here. All of these were significant in the discovery sample. All preregistered tests and hypotheses are listed in the methods. Additional control analyses including medication status as additional control variable can be found in Table E in S1 Text.

However, due to a between subject indexing error in the RT analysis in the discovery sample, the results reporting relationships between RT effects and transdiagnostic dimensions were invalid. We therefore did not attempt to repeat these analyses. None of the other analyses were affected, and the decision data and self-report data by themselves were sufficient to confirm and disconfirm our 3 main hypotheses. In each case, the hypotheses had been framed in terms of decision effects (i.e., in terms of which choices were taken as opposed to how quickly a choice was taken) and differing self-reported insights. The RT effects had only served as secondary evidence suggesting RTs were sensitive to the same factors as decisions.

## Results of testing hypothesis 1: Apathy is linked to decision inertia

Our first hypothesis was that apathy symptoms relate to increased decision inertia, i.e., being "stuck in a rut." We measured this using a computational model that gave us for each participant a measure of how likely to search again depending on how often they had already searched, beyond what they should do given the value, cost, and number of opportunities (measured as in Fig 2A, parameter estimate for "# Prev Searches," see also Fig 1F). Indeed, we found that apathy increased decision inertia (Fig 4A, mean = 0.066, 1-sided 95% CI$_{lower}$: 0.005, hypothesis 1A; for all regression analyses relating behaviour to clinical factors, see S4 Fig for corresponding correlation analyses). In other words, participants with more apathy symptoms were more likely to search again the more they had already searched. To understand this effect in more detail, we wanted to exclude other possible explanations for the decision inertia (S5 Fig) by linking measures of several of these alternative effects to apathy. First, participants

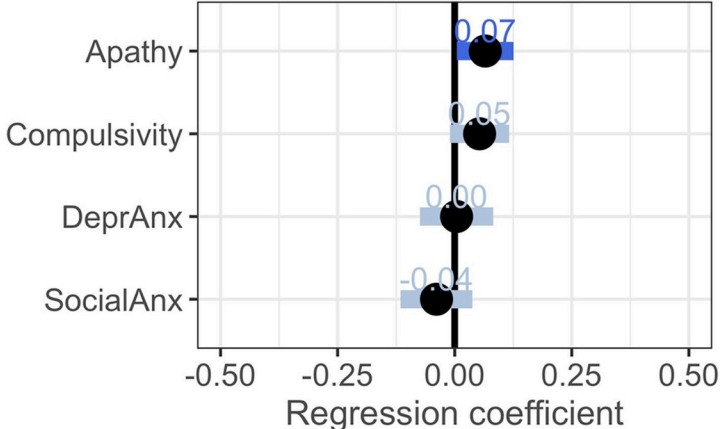

**Fig 4. Apathy symptoms and habitual overpersevering (confirmation sample).** To test whether higher scores on the apathy factor related to decision inertia ("stuck in a rut" bias), we did a regression analysis predicting choices of the bias based on the psychiatric factor scores, controlling for age, gender, education, and general performance on the task. The behavioural measures were derived from the analyses in Fig 2A, i.e., each person's computational model weight for the impact of the previous number of searches already done in a trial on choice (on the group level making people more likely to repeat a choice). **(A)** Higher apathy scores related to increased decision inertia, i.e., the higher the apathy score, the more participants were likely to search again the more they had already searched. Error bars show 95% Bayesian credible intervals (1 tailed). Significance is shown by credible intervals not including 0 and dark blue colour. Data in this figure are available in S9 and S16 Data.

might be overoptimistic about how likely they were to receive a desired outcome (Fig 2A, parameter "ProspVS1-Adapted"; see also S1 Text "Carrying out a choice sequence" for details of how this was computed in our model). We found no evidence in the choice data that apathy led to reduced downward updating of subjective estimates of prospective value as it objectively declined during the course of each sequence (mean = 0.014, 95% CI [−0.062 to 0.089])). Second, we also excluded the possibility that accumulated costs (or "sunk cost fallacy") could drive their behaviour (Fig 2A, parameter "Total cos"; see also S1 Text for details of computation). Finally, we also considered whether apathy might relate to not taking costs into account while going through a sequence of searches (Fig 2A, parameter "Cost"). We found no evidence for this in their choices (mean = 0.004, 95% CI [−0.072 to 0.083]).

### Results of testing hypothesis 2: Compulsivity and oversearching due to cost insensitivity

Our second hypothesis was that compulsivity would also relate to searching too often once participants started a sequence of searches, but that it would do so via a different mechanism than the one operating in apathy. Specifically, we hypothesised that it would be due to reduced sensitivity to costs (Fig 2A, parameter "Cost"; see also S1 Text "Carrying out a choice sequence" for model details). This had been the case in the discovery sample: Higher compulsivity was linked to taking costs into account less. In the confirmation sample, there was a similar effect but it fell just short of the threshold for significance (Fig 5A, mean = 0.061, 1-sided 95 % $\text{CI}_{\text{lower}}$: −0.006, hypothesis 2A). However, the effect was of a similar size in both discovery and confirmation samples; notably, the estimated mean of the confirmation sample was within the 95% credible interval of the discovery sample [38,39]. In other words, the mean of the confirmation sample lay within the range of expected population means that had been

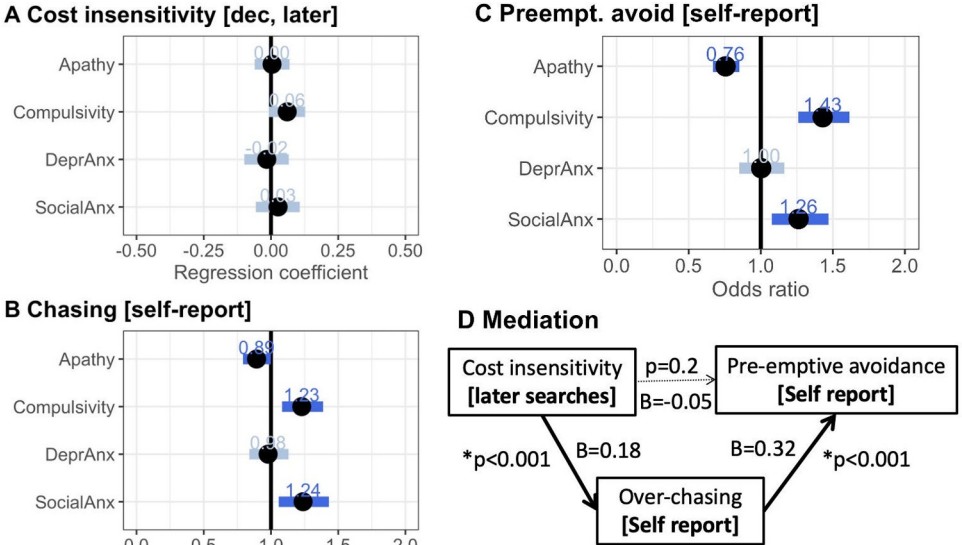

**Fig 5. Compulsivity scores, cost insensitivity during sequences of choices, and preemptive avoidance of biases (confirmation sample).** During later searches, higher compulsivity scores related to insensitivity to costs **(A)**. Compulsivity was further related to increased self-reported chasing **(B)**. **(C)** Increased compulsivity scores were, however, also related to increased self-reported preemptive avoidance of trials, which participants perceived as likely to lead to oversearching. **(D)** A mediation analysis revealed that the cost insensitivity bias during a sequence of choices was related (full mediation) to self-reported preemptive avoidance via self-reported awareness of a tendency to overchase. Stats in A–C are 95% Bayesian credible intervals (1 tailed); significance is shown by credible intervals not including zero and dark blue colour. D reports results of frequentist *t* tests at $p < 0.05$. Data in this figure are available in S2, S9, and S16 Data.

established in the discovery sample, suggesting a similar pattern of behaviour exists in both discovery and confirmation samples.

In addition to this somewhat ambiguous result, we had a complementary self-report measure of "overchasing" (Q2), i.e., an inability to stop searching because the alternatives appear attractive (even in the face of costs that lead to search termination in other participants). Here, we found that higher compulsivity scores were consistently related to more self-reported "overchasing" (Fig 5B, odds ratio = 1.224, 1-sided 95 % $CI_{lower}$:1.081, hypothesis 2B). Also see exploratory results for an association between self-reported cost insensitivity and compulsivity, based on an additional self-report measure included in the confirmation sample.

## Results of testing hypothesis 3: Compulsivity is linked to attempts to preemptively avoid situations in which sequential biases could arise

One rational response to knowing that one is biased when carrying out a sequence of searches would be to preemptively avoid situations where that bias is particularly likely to occur. We tested whether higher compulsivity scores were related to this preemptive avoidance and found this to be the case in terms of self-reported preemptive avoidance (Q3). We observed that compulsivity was linked to self-reported preemptive avoidance (Fig 4D, odds ratio = 1.423, 1-sided 95 % $CI_{lower}$:1.255, hypothesis 3C). One possible interpretation is that these 2 behaviours are linked by awareness of the bias; in other words, people consciously avoid situations that can lead to the emergence of their latent bias if they are aware of the bias. We tested this using a mediation analysis. We found that, indeed, awareness of "chasing" mediated the relationship between reduced cost sensitivity during a sequence of searches and self-reported preemptive avoidance (Fig 4E, indirect path: cost insensitivity [later searches] to overchasing [self-report] B = 0.18, z = 5.06, *p* < 0.001 and overchasing [self-report] to preemptive avoidance [self-report] B = 0.32,

z = 9.09, $p < 0.001$; but cost insensitivity [later searches] not directly to preemptive avoidance [self-report] B = −0.05, z = −1.29, $p = 0.20$, hypothesis 3C).

## Additional control analyses

We collected medication status in the confirmation sample (Table 1) to exclude it as another potential confound and found no change in the clinical associations reported above (Table E in S1 Text).

## Exploratory analyses—Predicting real-life individual differences through task behaviour and self-report

In addition to our prespecified tests, we wanted to know whether our cognitive task and self-reports of task behaviour, as a whole, had predictive power over the clinical and nonclinical demographic measures (Table 3). For this, we tested how well individual differences in the confirmation sample (in e.g., compulsivity) could be predicted from a model linking these measures to task/self-report measures in the discovery sample (see Exploratory Methods for details).

We found that compulsivity, age, gender and education, but not depression/anxiety or social anxiety, could be significantly predicted based on task behaviour and self-reports (Table 3, column "Significance in confirmation sample and correlation/neg LL").

Apathy was only associated with a single significant measure in the discovery sample, described above, and therefore was not tested again here.

We noted that predictions based mainly on RT measures were least likely to replicate (Table 3, column "Confirmation sample mean and Bayesian 95% CI (1 tailed)").

## Exploratory analyses—Measuring cost insensitivity through self-report

We also collected a direct self-report question of cost insensitivity (S2D Fig), on top of the related over chasing and task behaviour measure, to see whether this is also linked to compulsivity. As we expected, we found a strong link between the 2 (odds ratio = 1.292, 95% 2-sided CI [1.117 to 1.497]), further supporting our findings of compulsivity being related to changes in how costs are taken into account.

**Table 3. Predicting transdiagnostic dimensions and demographics (confirmation sample).**

| Clinical/demographic factor | Significance in confirmation sample and correlation/neg LL | Behavioural measure | Discovery sample mean and Bayesian 95% CI (2 tailed) | Confirmation sample mean and Bayesian 95% CI (1 tailed) |
|---|---|---|---|---|
| Apathy | N/A | PrevSearch [later dec] | 0.13 (0.03 to 0.23) | 0.07 (0.01 to 0.13) |
| Compulsivity | $p < 0.0001$, $r = 0.195$ | Cost [later dec] | 0.15 (0.05 to 0.26) | 0.06 (−0.01 to 0.13) |
| | | Offer [later RT] | −0.11 (−0.21 to −0.01) | 0.02 (−0.05 to 0.09) |
| | | Props Val S1 [later RT] | 0.10 (0.00 to 0.20) | −0.02 (−0.09 to 0.05) |
| | | Myopic [later RT] | 0.13 (0.02 to 0.24) | −0.03 (−0.10 to 0.04) |
| | | invTemp [initial dec] | −0.1 (−0.21 to −0.00) | −0.21 (−0.27 to −0.14) |
| | | Preempt Avoid [self rep] | Odds ratio 1.46 (1.20 to 1.79) | Odds ratio 1.42, CI90% (1.26 to 1.61) |
| | | Chasing [self rep] | Odds ratio 1.32 (1.10 to 1.59) | Odds ratio 1.22 (1.08 to 1.38) |
| DeprAnx | $p = 0.45$, $r = 0.005$ | DecrThres [self rep] | Odds ratio 1.35 (1.09 to 1.69) | N/A |
| | | Offer [later RT] | 0.13 (0.02 to 0.23) | −0.02 (−0.09 to 0.05) |
| | | Cost [later RT] | 0.10 (0.03 to 0.18) | −0.06 (−0.13 to 0.02) |
| | | Myopic [later RT] | −0.11 (−0.21 to −0.01) | 0.03 (−0.05 to 0.10) |
| SocialAnx | $p = 0.25$, $r = 0.027$ | RT.main [later RT] | 0.12 (0.01 to 0.23) | −0.07 (−0.15 to 0.01) |
| | | RT.main [initial RT] | 0.13 (0.01 to 0.25) | 0.01 (−0.07 to 0.10) |

(*Continued*)

**Table 3.** (Continued)

| Clinical/ demographic factor | Significance in confirmation sample and correlation/neg LL | Behavioural measure | Discovery sample mean and Bayesian 95% CI (2 tailed) | Confirmation sample mean and Bayesian 95% CI (1 tailed) |
|---|---|---|---|---|
| Age | $p = 0.0034$ $r = 0.104$ | Cost [later dec] | 0.11 (0.01 to 0.21) | 0.09 (0.02 to 0.15) |
| | | CostXPrevSearch [later dec] | 0.11(0.01 to 0.21) | 0.08 (0.01 to 0.14) |
| | | Offer [initial dec] | −0.12 (−0.23 to −0.02) | −0.13 (−0.19 to −0.06) |
| | | Myopic [initial dec] | 0.10 (0.01 to 0.20) | 0.12 (0.05 to 0.18) |
| | | Forage bias [initial dec] | −0.12 (−0.22 to −0.02) | −0.06 (−0.13 to 0.00) |
| | | Cost [later RT] | −0.13 (−0.24 to −0.03) | 0.02 (−0.05 to 0.08) |
| | | RT.main [later RT] | 0.34 (0.25 to 0.43) | 0.03 (−0.03 to 0.09) |
| | | RT.noise [later RT] | −0.16 (−0.27 to −0.05) | −0.07 (−0.13 to 0.01) |
| | | Forage Choice [later RT] | 0.15 (0.04to 0.26) | 0.03 (−0.03 to 0.10) |
| | | Preempt Avoid [initial RT] | 0.16 (0.05 to 0.27) | −0.03 (−0.09 to 0.04) |
| | | PreemptAvoid [self rep] | Odds ratio 0.82 (0.67 to 0.99) | Odds ratio 0.79 (0.70 to 0.89) |
| Gender | $p = 0.0253$ negLL = 499 | CostXPrevSearch [later dec] | 0.23 (0.03 to 0.44) | 0.15 (0.01 to 0.28) |
| | | RT.main [later RT] | −0.46 (−0.65 to −0.27) | −0.05 (−0.18 to 0.08) |
| | | RT.noise [later RT] | 0.39 (0.19 to 0.59) | 0.15 (0.02 to 0.28) |
| | | Inv temp [initial Dec] | 0.39 (0.18 to 0.60) | 0.25 (0.12 to 0.38) |
| | | RT.noise [initial RT] | 0.45 (0.25 to 0.64) | 0.03 (−0.10 to 0.17) |
| | | Offer [initial RT] | 0.27 (0.10 to 0.45) | 0.05 (−0.06 to 0.17) |
| | | Cost [initial RT] | 0.15 (0.01 to 0.28) | 0.04 (−0.07 to 0.16) |
| | | Myopic [initial RT] | −0.23 (−0.40 to −0.05) | −0.05 (−0.17 to 0.07) |
| | | Plan ahead [self rep] | Odds ratio 1.75 (1.13 to 2.69) | N/A |
| Education | $p = 0.0353$ $r = 0.065$ | ProspValDiff [later dec] | −0.13 (−0.26 to −0.03) | −0.09 (−0.16 to −0.02) |
| | | Forage Bias [later dec] | 0.17 (0.02 to 0.30) | 0.00 (−0.07 to 0.08) |
| | | RT.noise [later RT] | 0.14 (0.03 to 0.29) | 0.05 (−0.02 to 0.12) |

Regression models fitted on the discovery sample data and predicting transdiagnostic and demographic measures were applied to the confirmation sample. Significance of the predictions was assessed either using correlations between predictions and measured data (all variables other than gender) or using "neg LL" as measure of goodness of model fit (gender). In summary, compulsivity was the factor we could predict best from a mixture of self-report and behaviour. While we could also predict, age, gender, and education, the included behavioural/self-report measures are listed separately for the discovery sample (95% 2-tailed Bayesian CI, based on analyses predicting the behavioural/self-report measures on all clinical and demographic measures) and the confirmation sample (95% 1-tailed CIs); measures that replicated are highlighted in grey. The self-report factor DecrThres ("Would you accept offers you would have previously rejected?") was not part of any preregistered hypotheses and thus accidently not collected in the confirmation sample, which is why it could not be used in the confirmation sample. See Tables F–H in S1 Text for full list of all links between clinical/demographics measures and task behaviour and self-reports. Data in this table are available in S2, S9–S12, and S16 Data.

CI, credible interval; neg LL, negative log likelihood; RT, reaction time.

## Discussion

### Summary

In natural environments, decisions are not made in isolation from one another. Instead for many active decision-making agents, whether human or animal, the choices taken at one point in time may have implications for which opportunities may be encountered later [7,10,13]. We were interested in how participants flexibly plan and adjust sequences of decisions and how

this relates to different psychiatrically relevant dimensions in a large confirmation sample of participants ($n$ = 756). We preregistered analyses based on an initial discovery sample ($n$ = 449) and found a high degree of reproducibility, i.e., 4 out of the 5 tests supporting our 3 hypotheses fully replicated. Specifically, we found that both apathy and compulsivity related to failures to stop searching appropriately, but for different reasons. Apathy was linked to "decision inertia," i.e., a tendency to search again and again for a new offer, especially when the participant had searched several times before, regardless of whether this was appropriate. In contrast, compulsivity was linked to changes in how people thought about costs; while the evidence of increased insensitivity to costs in choices was somewhat ambiguous, there was clear evidence of changes in the perception of costs both when directly asked about cost insensitivity and as a sense of "overchasing," i.e., the feeling of going after high value options longer than they should. Individuals who scored highly on compulsivity also reported taking preemptive action to avoid situations in which their bias would have affected their decision-making. Our results suggest that apathy and compulsivity are linked to distinct biases in sequential search.

## Transdiagnostic approach, ecological paradigm, and preregistration

Traditionally, psychiatric research has thought of clinical disorders as defined by a set of specific symptoms. However, more recently, researchers have begun to look at transdiagnostic dimensions of symptoms, cutting through traditional disorder boundaries [15]. In this framework, dimensions such as apathy or compulsivity emerge. Importantly, these dimensions exist on a spectrum from health to disease.

Teasing apart the role of distinct yet partially correlated transdiagnostic symptoms was made possible here by testing a large number of participants over the internet. For this approach to work well, it is necessary that symptom scores span the range from healthy to psychiatrically relevant. We found here, as have others previously [37,40,41], that participants in an internet based study showed just such a range of symptoms and in fact even showed symptoms at higher rates than found in the normal population. To ensure that the self-reported symptoms were valid, we included several quality checks. Previous work [20] that directly compared transdiagnostic dimensions derived from self-reports over the internet to clinician rated diagnoses found good correspondence between them and crucially also that dimensions related more clearly to behaviour than diagnostic labels.

We used a sequential search task as many decisions in real life involve planning a sequence of decisions and carrying them out while monitoring whether the preplanned strategy is still appropriate. Such decisions are often made between a current offer and the opportunity to carry on searching for even better offers. This contrasts with more commonly studied tasks involving single decisions between specific well-defined offers. We and others have shown that these processes are neurally and behaviourally distinct [4,8,42]. We used a previously established computational model [14] to tease apart the different component cognitive processes underlying search behaviour, and in this way, we identified different types of sequential bias. We complemented this approach with analysis of participants' self-reports of their own behaviour to test their metacognition. We ensured good data quality by giving participants automated training on the task and testing their knowledge using a multiple choice test they had to pass before being admitted to the main study. As a consequence, data quality was very high (only few participants had to be excluded based on task performance).

Many of our results highlight the usefulness of acquiring reports of subjective experience related to the task performed (see also [43]): Participants themselves might not have direct access to the decision model (including biases) they implement. Instead, they make observations about their own behaviour. Understanding such metacognition of decision-making

preferences and biases further complements our understanding of the cognitive processes during decisions based on cognitive models of the task behaviour itself. If we want to positively affect people's behaviours, understanding how people think about what they are doing could be of great benefit. In particular, understanding the interactions of cognitive models with experience, behavioural strategies, and metacognition, as we have done here, could help the development of better cognitive approaches to improve behaviours long term.

In this study, we took a 2-step discovery and confirmation sample approach to combine statistical rigour with the possibility that the data themselves might help inform our hypotheses. We preregistered the study based on the results of the discovery sample. The ability to collect very large samples of participants over the internet with ease allowed us to achieve high power (95%) even for associations between clinical factors and task behaviour of a small effect size (correlations $\sim r = 0.1$). Overall, we found that the evidence relating to our 3 primary hypotheses was reproducible. In the future, it might be worthwhile considering performing power calculations assuming a smaller effect size than found in a discovery sample [39].

## Apathy

The dimension we termed "apathy" pulled together measures from different questionnaires related to emotional and social apathy, anhedonia and externally oriented thinking (from alexithymia questionnaire [44], example item: "I prefer just to let things happen rather than to understand why it turned out that way") in agreement with previous studies that have highlighted strong conceptual and empirical overlap between the constructs of apathy and anhedonia [19]. Of note, we found that the apathy dimension was distinct from another dimension capturing other symptoms of depression, anxiety, and fatigue, in agreement with previous findings in neurodegenerative disorders [45], Parkinson disease [46], depression [47], and healthy volunteers [35]. In contrast, however, behavioural apathy appeared distinct and was related to the depression/anxiety dimensions. This suggests that apathy is multifaceted.

We found that the higher participants scored on our dimension of apathy the more decision inertia they showed. Our results highlight how apathy can counterintuitively lead to the exertion of more unnecessary effort (i.e., searching too often). Unnecessary effort exertion could potentially lead to further exhaustion and apathy and, therefore, as a result of a vicious cycle, to the exertion of more and more unnecessary effort. Within work psychology, "overcommitment," defined as an inability to disengage from work has been highlighted as an intrinsic factor modulating the impact of the working environment [24,48,49]. Our findings are also in broad agreement with a previous report in a small sample of patients with depression, which is often comorbid with apathy, of delayed stopping in the "secretary problem" task [50]. We also note that we did not find an effect of apathy on general response slowing at the initial decision. This was potentially the case because our task had a fixed "monitoring" period (3 to 6 seconds) on each trial to ensure participants considered all information before responding. However, this may have reduced sensitivity to individual differences in RT.

Our findings may initially seem at odds with other reports that apathy or anhedonia lead to avoidance of effort when participants choose between 2 options that vary in effort and reward [51–53]. However, importantly, participants make a very different kind of decision in our paradigm—they make sequences of decisions. This means that at each choice point, previous decision plans can be retrieved and potentially adjusted, i.e., the decision can be reconsidered. Thus, thinking about stopping, is in itself, a deliberate cognitive act, which is affected by apathy. This interpretation would be in agreement with previous clinical reports that patients with apathy show normal behaviour if externally prompted, but struggle to self-generate behaviour [22], as well as experimental findings of reduced generation of new options as candidates for decision-making [23].

## Compulsivity

The dimension we called "compulsivity" included different subscales of the OCI-R [54], with the notable exception of the obsessive thinking subscale, as well as the unusual experience subscale of the schizotypy questionnaire (short scales for Measuring Schizotypy [55]). We did not include broader questions measuring compulsivity in the context of other disorders such as eating disorders or gambling. Compulsivity was primarily related to changes in self-report.

Compulsivity led to self-reported "overchasing", i.e., the subjective experiences of searching for too long in the hope of getting a better offer, which, in turn, was linked to a reported tendency to preemptively avoid even beginning to explore potentially costly courses of action. This interpretation was bolstered by a finding of clear changes in self-reported cost insensitivity in an exploratory analysis. However, we found that objective measurements of cost sensitivity per se were less clearly related to compulsivity. While they were correlated with compulsivity in the discovery sample, such an association was not quite significant in the confirmation sample. However, on the other hand, it is equally important to note that the confirmation sample mean fell within the Bayesian credible interval of possible population means of the discovery sample; such a finding is often taken as evidence of confirmation of a result [38,39]. Importantly, using a higher inclusion threshold for the effect in the discovery sample, such as $p < 0.005$ as has been suggested [56], would not have changed this result, as the original effect was highly significant.

Our self-report changes of overchasing and cost insensitivity are also in agreement with the clinical profile of OCD patients carrying out compulsions despite their immense time costs [28]. It is also related to experimental findings of reduced cost sensitivity in the context of oversampling information before making a decision [57,58]. Our findings are also in broad agreement with previous reports linking compulsivity to an increase of model-free (or "habitual") mechanisms controlling behaviour rather than model-based (or "goal-directed") mechanisms [59–61]. What our findings suggest is that the reason that control might change from model-based to model-free is because of an underestimation of the costs associated with the model-free behaviour (i.e., in our case the sequential search costs). While the most commonly used psychological treatment, cognitive behavioural therapy is based on the assumption that obsessions are primary [25], this has been challenged. Specifically, while there are patients with compulsions without obsessions [29], the reverse is rarely the case [30]. If obsessions were primary to compulsions, one would expect compulsions only to emerge after specific obsessions have developed. Our findings align more with the view that compulsions can be a primary deficit. This is in agreement with the ego-dystonic nature of OCD and recent findings of intact metacognition in OCD, but failure to use this knowledge to behave appropriately [62]. However, whether OCD leaves metacognitive abilities completely intact is unclear; a study using a transdiagnostic, large-scale approach, instead of a case–control approach, suggested compulsivity can affect both metacognition and behaviour [40]. In fact, worse metacognitive ability and overconfidence in one's ability have also been found in other paradigms [63,64], suggesting that metacognitive processes and behaviour are not completely independent.

Finally, our data suggest that people perceive themselves to be implementing preemptive avoidance strategies, i.e., try to avoid situations where they fear they will behave suboptimally in the future, as a function of their compulsive symptoms. Importantly, we found that the cost insensitivity bias did not directly drive the appearance of a preemptive strategy. Instead, the subjective experience of overchasing for better options (despite the costs being incurred) fully mediated the relationship between the cost insensitivity bias and self-reported preemptive behaviour. These findings align with clinical reports that OCD patients avoid situations that provoke their symptoms by, e.g., asking family members to wash their clothes for them [32].

Of particular note in our data is that such avoidance can be related to the behavioural bias per se, rather than a specific obsession. This is again in contrast to a view that obsessions are primary in OCD and that patients avoid situations because they trigger their obsessions, which then, in turn, trigger their compulsions.

While we find that compulsivity is linked to increased self-reported preemptive avoidance, there was no such link to a corresponding behavioural measure aimed at this (Table G in S1 Text). This could suggest either that participants only subjectively perceive a change in their behaviour, which is in fact unchanged, or it could suggest that the behavioural measure does not capture the full range of behaviours participants might engage in order to preemptively avoid but which they consider when they self-report.

### Predicting individual differences with behaviour and self-reports

In exploratory analyses, we used a data mining/prediction approach to build models that used all potentially informative measures together to predict participants' transdiagnostic and demographic features similarly to [65]. These models were built based on the discovery sample, taking advantage of all significant associations in the data. We tested predictive performance in the confirmation sample. We found that compulsivity, age, gender, and education could be predicted from a combination of task behaviour and self-reports. In contrast, depression/anxiety and social anxiety could not be predicted. Note that apathy was not included in this analysis as there was only one significant association in the discovery sample, i.e., the link with decision inertia described above, meaning no model using a combination of measures for its prediction could improve predictive power.

Interestingly, within this exploratory analysis, we noticed that the least replicable links between task behaviour and clinical/demographic features were related to RTs. This could have been because our online setup was not optimised for RT collection, e.g., we did not impose response deadlines or reward speeded responding.

Overall, the analysis highlights the possibility that links between metacognition, task behaviour, and clinical dimensions can be multifaceted, i.e., not map one to one with each other. In our data, this was the case for compulsivity, which was best predicted by a mixture of metacognitive measures (e.g., self-reported chasing) and behavioural measures (such as choice stochasticity). Importantly, to find and validate such multifaceted relationships, a discovery confirmation sample approach is particularly valuable as it allows for the initial identification of the combination of the relevant factors in the discovery sample and a test of these factors in the confirmation sample. While we had no prior hypotheses about age and gender, the mixtures of effects are in line with previous studies showing differences in risk taking between male and female participants [65] and changes in decision-making with age [66]. However, again, they highlight how with complex traits, a larger set of behavioural and metacognitive changes can emerge in more complex tasks.

### Summary

In conclusion, during complex search sequences, behavioural biases emerged in a way that was related to participants' tendencies towards compulsivity and apathy. Combining our clinical dimensions with our computational model and self-reports of participants' own task experiences, we could further specify how participants' behaviour, experience, and behavioural strategies interact and how those relationships link to differences in their clinical profile.

### Methods

Deviations from the Stage 1 protocol are indicated by superscript letters; these deviations can be found in the section "Protocol deviations." To enable both data-informed hypothesis

generation and testing as well as statistical rigour, we followed a 2-step discovery and confirmation sample procedure. First, we collected a discovery sample, which we analysed and used to define a precise set of hypotheses linking clinical dimensions with our task behaviour. This was accepted as a Stage 1 Preregistered Research Article. See the OSF link here: https://osf.io/dfg2u. The results of the discovery sample together with the preregistered hypotheses are all contained in the preregistration. Hypotheses of interest that were preregistered concerned relationships between clinical questionnaires and task behaviour. These are 1 tailed. For any statistical tests establishing group average effects of behaviour, we used 2-tailed tests as they were not included in the preregistered hypotheses.[a]

## Participants

Participants were recruited via the online platform Prolific.ac. We aimed to acquire 750 data sets that fulfilled the preregistered inclusion criteria detailed below; 756 were collected instead due to technical reasons in the Prolific system. This sample size was determined based on the discovery sample shown below to give a power of 0.95 (see Fig 6 and section "Power calculation" below for details). Ethical approval for the study was given by the Oxford University Central University Research Ethics Committee (CUREC) (Ref-number: R54722/RE001). All participants provided informed consent before taking part in the study.

## Task

In a computerised foraging task, participants made repeated choices with the aim to earn as much money as possible (Fig 1C). On each of 150 trials, participants were shown one option that we refer to as the "default offer" (point range 5 to 150 in steps of 5). The offer was the default option that the participants would have unless they decided to spin a "wheel of fortune." The alternatives on the wheel differed in their magnitudes (same range as initial offer)

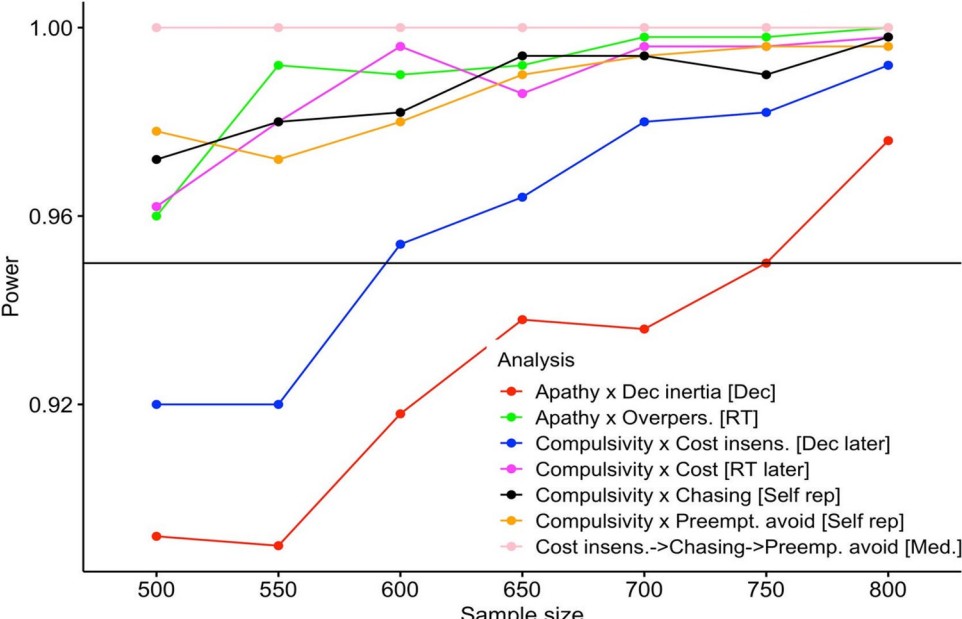

**Fig 6. Power analysis.** Power increases with sample size. To reach 95% power in the analysis with lowest power (i.e., analysis of the impact of apathy on overperseverance measured in the participants' decisions), we selected 750 participants for the confirmation sample.[n] RT, reaction time.

and in their probability (i.e., the size of the area they occupied on the wheel). If participants decided to "bank" the offer, the trial was finished. If, however, they decided to spin the wheel, they first had to pay a cost (either 0, 7, or 15 points, fixed for the whole trial, but varied across trials) and then spun the wheel. They would get a new offer depending on where the wheel stopped. There were between 2 and 6 alternatives on the wheel on each trial. Participants were shown how often they could spin the wheel on each trial ("spins remaining"; varied between 1 and 10 times across trials); they had to pay the same cost for each spin.

Timings: There was a random delay of 3 to 6 seconds (uniform) between the start of a trial and when participants could make their choice (buttons were shown as "disabled") to ensure that participants thought carefully about their choice. If participants chose to spin the wheel, the wheel spun for a random duration of 0.3 to 2 seconds. There was no enforced delay for any choices after the first one on each trial. When a trial was finished, there was a constant delay of 1.5 seconds before participants could move on to the next trial.

Training: Participants were given standardised instructions and 5 training trials. After, they had to complete a multiple choice test. If they passed the test, they could move on the task; if they failed it, they had to repeat the instructions and the multiple choice test.

Design of task schedule: We constructed 10 random task schedules that fulfilled the following criteria (see S1 Text, section "Task schedule design and model validation" and S7 Fig and Table A in S1 Text for details): First, correlations between variables influencing decisions (see analyses below for details) were below 0.7. Second, parameter recovery showed Pearson correlations of $r > 0.85$ between simulated and recovered parameters for all parameters. Third, parameter recovery across all schedules showed $r > 0.85$ for all parameters, suggesting that parameter estimates obtained using different schedules were comparable i.e., there was no systemic schedule bias affecting the results. Any given participant shares the schedule they performed with a tenth of the other participants.

The study was programmed in JavaScript and the study was run using JATOS [67].

## Questionnaires

After the task, participants completed a selection of questionnaires to assess a range of psychiatric symptoms: AMI [68], OCI-R [54], Beck Depression Inventory (BDI) [69], Spielberger State Anxiety [70], Liebowitz Social anxiety [71], Short Scales for Measuring Schizotypy [55], ("unusual experiences" and "introvertive anhedonia" subscale), Fatigue scale [72], and Toronto alexithymia scale [44]. Before completing the questionnaires, participants were shown a message highlighting the importance of the questionnaires for the study and were told that some questions might repeat and asked to answer as consistently as possible. Five questions were repeated to screen for data quality (see below). If participants responded faster than the fastest quartile of participants from the discovery sample with included data, they were shown a warning and asked whether they wanted to check their answers again.[b]

We also included several questions for participants to report their metacognition (i.e., insight) into how they behaved during the task. As question related to insight about decision inertia, we asked participants, "Did you ever feel like you continued spinning the wheel just because you had spun the wheel already this round, rather than because you really thought this was the right thing to do?" (Q1) For the question related to overchasing, we asked, "Did you ever feel like you spun too many times in a round because there was an option on the wheel that you really wanted to get?" (Q2) For the question related to preemptive avoidance, we asked, "Did you ever avoid starting spinning the wheel in the first place in a round because you were worried that you might end up spinning it too many times?" (Q3). All self-reports were presented as a scale from "never" to "always" in 7 steps.

To obtain additional control covariates for analyses, we also asked participants about their age, gender, years of education (General Certificate of Secondary Education [GCSE] A-levels/ High school, BSc, MSc, and PhD) and whether they were currently taking any psychoactive medication or drugs (which will be coded as distinct categories whenever drugs listed by at least 10 participants fell in a category, e.g., serotonergic, noradrenergic, dopaminergic, GABA-ergic, and "other"; otherwise just listed as "psychoactive medication yes/no") and, if applicable, frequency and dose. It turned out that in the confirmation sample, responses to specific medications were too infrequent and therefore medication was coded as "yes/no."

## Participant exclusion criteria

As our data were acquired online, we took several steps to ensure data quality. First, we ensured participants fully understood the task instruction by giving them an automated tutorial followed by a multiple choice test that they needed to pass with 100% correct before being able to do the task. Second, we gave participants bonus payments depending on their performance to increase motivation. Third, we excluded participants based on task performance as well as poor completion of the questionnaires. We excluded participants whose behaviour either suggested they did not engage with the task or whose responses were too unbalanced to allow analysis. Behavioural analyses were performed separately for the initial search decision on each trial and for all later searches, see below, and data exclusion reflected this. Participants were excluded due to bad performance if, when considering only the initial choice of each trial, for the 10 trials with highest offer minus search value (i.e., strongest value pressure to accept) they accepted the initial offer less often than for the 10 trials with the lowest offer minus search value. Participants were excluded due to imbalanced responses if—separately for the initial search decision of each trial and for later searches—they had fewer than 25 accept or spin responses.

Exclusion based on questionnaire responses:

1. Not completing all psychiatric questionnaires

2. Five questions were repeated. If a participant's responses were different by 2 or more points on average, then they were excluded.

Participants who only failed inclusion criteria for the behavioural models were included in the factor analyses of the questionnaire data (see below). Participants who only failed inclusion criteria for one behavioural model (e.g., only for the initial search) were included in the analyses of the questionnaire data and the other behavioural model.

## Data analysis

Data analysis was performed in MATLAB and R (version 4.0.4), including the packages rstan [73] (version 2.21.2), shinystan [74] (version 2.5.0), brms [75–78] (version 2.15.0), psych [79] (version 2.1.3), processR$^c$ (based on PROCESS [80], version 0.2.6) implemented using lavaan [81] (version 0.6–8), ggplot2 [82] (version 3.3.3), ggpubr [83] (version 0.4.0), tidyverse [84] (version 1.3.1), and sjPlot [85] (version 2.8.7).

## Computational model of prospective value

We used a previously described decision tree model to determine the value of searching for each trial [14]. In short, this model took into account the possibility of searching more than once on a given trial (up to the maximum number of searches available), the magnitudes and probabilities of the alternatives and the cost of searching. See S1 Text ("Decision tree model to

derive prospective value") for details. We illustrate the key factors contributing to our model-based prospective value estimate (see Fig 1D). For this, we first conducted a regression predicting prospective value based on the average value of all alternative options in the environment (myopic value), the standard deviation of all alternatives, the number of searches available, the cost of searching, and all possible 2- to 4-way interactions. To compute how well these components together approach prospective value, we computed the variance explained ($R^2$).

## Analysis—Decisions

We analysed choices to accept an offer or search with a decision model. In the model, we predicted decisions based on a linear combination of the different factors that should drive choices. We analysed the initial search decision on each trial and the subsequent searches separately. This means that we allow for differences in the processes underlying the decisions. This may be the case because for the initial decision, participants had time (forced delay of 3 to 6 seconds) to monitor the options and plan ahead (and we found previously that indeed they planned ahead [14]). In contrast, later decision could be executed immediately and could have been preplanned prior to their onset.

In the model of the initial search, we included as factors as previously [14]: myopic value (i.e., the average of the alternatives), prospective value (see above), the offer value, the cost of searching, and how much prospective value might change over the course of a trial. The last factor captures a measure for each person to what extent they preemptively avoided trials in which sequential bias would be more costly (as the faster prospective value might decrease, the more costly it could be to oversearch). All individual effect sizes of factors influencing participants' decisions can subsequently be related to clinical dimensions and introspective measures.

Specifically, we first computed the utility of searching versus accepting the offer ("Decision model—first search"):

$$Utility = b_{SearchBias} + b_{Offer} * Offer + b_{Cost} * Cost + b_{Myopic} * Myopic + b_{Prospectve} * Prospective$$
$$+ b_{AvgProspVChange} * Average\ prospective\ value\ change,$$

where parameters (e.g., $b_{SearchBias}$) were allowed to vary for each person (see below). For computational reasons [86], parameters were standardised by the some of their absolutes:

$$k = abs(b_{SearchBiasPre}) + abs(-1) + abs(b_{CostPre}) + abs(b_{MyopicPre}) + abs(b_{ProspectivePre})$$
$$+ abs(b_{AvgProspVChangePre})$$

and then as an example,

$$b_{SearchBias} = \frac{b_{SearchBiasPre}}{k}$$

The variable "average change in prospective value" above captures how much the prospective value might change over the course of a trial, meaning that participants would need to adapt their behaviour more.

$$AvgProspValChange = \frac{ProspVal_{Search1} - ProspVal_{last\ search\ before\ PropcVal=0}}{Maximum\ number\ of\ searches}$$

The utility was then translated into a probability of choosing to search using a standard

soft-max decision rule:

$$p(Search) = \frac{1}{\varepsilon^{-\beta * Utility}},$$

where β was the inverse temperature, i.e., a measure of choice noisiness.

Note that this decision model is a reparameterised logistic regression analysis [14] with the main difference being that one single parameter (inverse temperature) is used to capture choice noisiness, rather than choice noisiness affecting the size of the estimates of all parameters (e.g., if a participant is noisy, in a regression all regression weights might appear smaller than for a less noisy participant, thus introducing strong correlations between the estimated parameters which hinders the goal of relating specific parameters, i.e., behavioural measures, to specific psychiatric factors). It also means all regression weights (bs) are relative weights, i.e., represent the relative contribution of that regressor on the decision.

The model was fitted as a hierarchical model using full Bayesian inference with MCMC sampling [73]; all variables were z-score normalised across all participants to ease fitting. In the hierarchical model, we specified weak priors for all group level parameters (normal distribution with mean 0 and standard deviation 3). The parameters for individual participants were then related to the group level distribution as [$parameter_i$ ~ normal($mean_{group}$, $stdev_{group}$)). For all parameter other than the inverse temperature, inspection of individual participants' parameters of the discovery sample (from a nonhierarchical version of the model) suggested that a normal distribution on the parameters was appropriate, while for the inverse temperature, a normal distribution was applied to log transformed data ($\log(invTemp_i)$ ~ normal($\log(invTemp_{group})$,$stdev_{group}$). Models were run for 4,000 samples per chain (of which 2,000 were warmup samples)[d], with 4 chains. Absence of divergent samples was verified using the shinystan diagnostic toolbox [74] and convergence of the model fit was assessed using the Rhat <1.1 criterion [87]. In the confirmation sample, significance of regression weights will be assessed using 1-tailed (2-tailed in the discovery sample) 95% Bayesian credible intervals.

We analysed the later search decisions on each trial to obtain different measures of sequential biases. To capture decision inertia, we included the previous number of searches on a trial. To capture sunk cost fallacy, we included the total costs incurred so far on a trial. To capture whether participants did not adapt their impression of how valuable it was to search on a given trial as fewer and fewer searches remained, we included how much the prospective value had changed since the start of the trial. To capture whether participants ignored the cost of searching in favour of chasing high valued option, we included the cost of searching ("Decision model—later searches"):

$$\begin{aligned}
Utility = {} & b_{SearchBias} + b_{Offer} * Offer + b_{Cost} * Cost + b_{Myopic} * Myopic + b_{ProspectiveVS1} \\
& * ProspectiveValueSearch1 + b_{ProspVS1-Adapted} * (ProspectiveValue\ Search1 \\
& - Current\ Prospective\ Value) + b_{PreviousSearches} * Previous\ number\ of\ searches + b_{TotalCost} \\
& * Total\ Cost,
\end{aligned}$$

where again the regression weights were regularised as described above, by dividing through k, i.e., the sum of their absolutes.[e]

We checked how well the behaviour of the fitted model resembled the behaviour of real participants (posterior predictive checks). For this, we extracted, for each participant and for each search decision, the model's probability of searching as opposed to accepting the offer, given the parameters of the best fitting model. These are shown in Fig 1E and 1F together with the data from real participants.

## Analysing behaviour—RTs

We analysed how different factors influenced participants' RTs using multiple regressions similar to the decision analyses to look at the factors influencing deliberation times. For the initial decision, we can look at which planning factors participants deliberate over during the initial planning. In the later decisions, we can look at the factors participants consider during execution and updating of the decision strategy as the number of opportunities depletes.

First, as there was no response deadline, outlier RTs (very slow RTs) were possible. RTs over 10 seconds were therefore discarded. After outlier rejection, RTs were modelled using a hierarchical regression with a shifted lognormal distribution, fit using the brms package that implements Bayesian multilevel models in R, using exactly the same regressors as for the decision model ("RT model—initial search"), shown here in standard regression modelling notation (as implemented in R):

$$RT \sim 1 + Offer + Cost + Myopic + Prospective + Average\ prospective\ value\ change + (1 \\ + Offer + Cost + Mypoic + Prospective + Average\ prospectiive\ value\ change\ |ID),$$

where "|ID" appears in Wilkinson notation and indicates that all regressors were given a "random" in addition to a "fixed" effect or, in other words, that they were modelled hierarchically. We also allowed decision-to-decision variability of the RTs to vary between participants (following a hierarchical distribution). All regressors were z-score normalised. We used the standard priors in brms for the group level (i.e., flat prior for group level regression weights for all parameters). Convergence was again assessed as described above. Models were run for 12,000 samples (of which 8,000 warmup samples) with 3 chains and within chain parallelisation.[f]

For the later searches ("RT model—later searches"), we again included exactly the same regressors as for the corresponding model of decisions.

$$RT \sim 1 + Offer + Cost + Myopic + Prospective\ Value\ Search\ 1 \\ + (Prospective\ Value\ Search\ 1 - Current\ Prospective\ Value) \\ + Previous\ number\ of\ searches + Total\ Cost + (1 + Offer + Cost \\ + Myopic + Prospectove\ Value\ Search\ 1 \\ + (Prospective\ Value\ Search\ 1 - Current\ Prospective\ Value) \\ + Previous\ number\ of\ searches + Total\ Cost\ |\ ID)$$

## Factor analysis of questionnaire measures

Summary: In the discovery sample, we performed an exploratory factor analysis. Here, this was validated using confirmatory and exploratory factor analysis. As this showed good replication of results, we next applied the factor weights from the discovery sample to the confirmation sample to obtain factor scores for each person for each factor.[g]

In detail: We performed exploratory factor analysis of the discovery sample to derive the transdiagnostic clinical dimensions within the psychiatric questionnaires using the R package "psych." The aim of factor analysis is to explain the relationship (correlation) between the questionnaire items or subscales as a function of underlying factors. For this, we computed the previously established subscale summary scores for each questionnaire and entered those scores into a factor analysis. We used summary scores rather than raw questionnaire responses because, first they have better properties (e.g., they have better reliability and better distribution shapes and are continuous rather than categorical, allowing Pearson correlations to be computed) [88]. Second, they offer a better ratio of number of participants to subscales than to items. We followed standard procedures with the factor analysis [89]: First, we obtained the correlation matrix of all subscales

(Spearman) as this is the data that factor analysis aims to explain. The second step was to determine how many factors were appropriate to describe our data. For this, we used parallel analysis [90]. Third, we performed a factor analysis, using generalised least square to fit the model. The final step involved "rotating" the factors (in a 2D coordinate system, one could visualise moving the 2 axes). This is done to improve the interpretability of the factors—unrotated, the results of factors analysis are such that items often show a strong relationship with several factors (high "crossloading"). After rotation, this is reduced, making the meaning of each factor easier to interpret, but allowing correlation between the factors. We used oblique rotation (oblimin) for this step. This revealed 4 factors in the discovery sample that we labelled "Apathy," "Compulsivity," "Depression/Anxiety," and "Social anxiety."

## Validation of factor analysis[h]

To validate the factor analysis in the confirmation sample, we ran 2 complementary checks. First, we performed a confirmatory factor analysis. For this, we used the Lavaan package [85], specifying 4 factors, as in the exploratory factor analysis on the discovery sample. Factors were allowed to correlate but subscales were assigned to a single factor (the one with the highest loading in the discovery sample). However, in reality, the loadings of some subscales were actually relatively evenly spread across factors, making the categorical assignment less than ideal (Fig 3, S3 Fig). We judged fit using 3 distinct indices, i.e., CFI, SRMR, and RMSEA.

As the results of the confirmatory factor analysis did not unambiguously suggest that the 4-factor solution was a good fit to the data, we followed our prespecified procedure and next performed an exploratory factor analysis on the confirmation sample. The same methods were followed as described above. We then visually compared the results of the factor analyses of discovery and confirmation sample. To quantify the similarity between the factor scores (for each person, for each factor) obtained if either weights of the discovery or of the confirmation sample were applied to the confirmation sample, we correlated them.

## Relating questionnaires to task performance and self-reports

We used the discovery sample to establish the exact links to be tested in the confirmation sample between questionnaire measurements, i.e., clinical dimensions and task performance (decisions, RTs, and post-task self-reports). Specifically, for the analyses of decisions and RTs, we performed linear regressions (R using Stan and brms software) in which the behavioural/self-report measures were related to the 4 psychiatric factors (compulsivity, apathy, depression, and social anxiety), while controlling for the covariates gender (coded as categorical variable with levels female, male, or "other"); education (coded as ordered factor [75] with levels: GCSE, A-levels/High school, BSc, MSc, and PhD; and age. In the analyses of parameters derived from the decision model, we included inverse temperature as a measure of general task performance and choice consistency. Similarly, in the analysis with parameters of the RT model, we included the intercept (as measure of general RT) and sigma (as measure of RT variability) from the model described above as covariate. In the analyses of the post-task self-reports, responses were treated as ordinal and monotonically increasing (rather than continuous) and therefore modelled with a cumulative distribution as implemented in brms [76]. All continuous regressors were z-score normalised. To illustrate the relationship between psychiatric factors and behaviour, we also plot partial correlations of the residuals. Residuals were computed in 2 regressions. First, the behaviour of interest was predicted as a function of all factors (i.e., psychological and control measures) listed above, apart from the one of interest (e.g., apathy). Second, the residual variance in the behaviour was correlated with the questionnaire dimension of (e.g., apathy) using nonparametric correlations (Kendall's tau) to ensure

results were not driven by outliers. For the analyses of the self-report questionnaire scores, residuals could only be computed by treating them as continuous.

The exact same procedure was applied to both discovery and confirmation sample. Significance in the confirmation sample was assessed using 1-tailed 95% Bayesian credible intervals (1 tailed, as the hypotheses were derived from the discovery sample). For the linear regressions weights (predicting behaviour based on clinical dimensions), Bayesian credible intervals excluding zero were judged as significant and confirming our hypotheses, while credible intervals including zero were judged as not significant. For the ordinal regressions (predicting self-reports based on task behaviour), Bayesian credible intervals on the parameter estimates (odds ratios) were judged as significant if they did not include one and they were judged as not significant if they included one.[i] We choose the 95% credible intervals as those are the intervals that determine the 95% likelihood of containing the true parameter value, given our data.

Mediation and moderation analyses (see below) were run using the R package processR (via R package lavaan) [80,91]. Significance was be determined as $p < 0.05$ 1 tailed.

From the discovery sample, we had derived specific compound hypotheses that were be tested in the confirmation sample (summarised in Table 3). For completeness, we also report all significant findings from the discovery sample again in the confirmation sample but marked as exploratory (see below).

We have removed all elements of the 3 hypotheses that were linked to RTs as an indexing error in the discovery sample that was used to define statistical test meant those were incorrect. We include the removed text explaining the RT elements in the footnotes below and show it in Table 4 as greyed out.[j]

**Hypothesis 1: Apathy is linked to decision inertia.**   To test this hypothesis, we used as dependent variable the measure of decision inertia from the decision model of the later searches (i.e., each person's parameter $b_{\#PrevSearches}$). In the discovery sample, the higher the apathy, the more participants tended to search again the more they have already searched (i.e., larger $b_{\#PrevSearches}$)—hypothesis 1A.[k]

**Hypothesis 2: Compulsivity is linked to a bias to oversearch due to cost insensitivity.** To test this hypothesis, we used as dependent variable the measure of cost sensitivity from the decision model of the later searches (i.e., each person's regression weight $b_{Cost}$). In the discovery sample, the higher the compulsivity score, the more a participant was insensitive to costs—hypothesis 2A. In the discovery sample, participants were aware of their bias, and, therefore, compulsivity was also linked to increased self-reported over chasing (self-report Q2)—hypothesis 2B.[l]

**Hypothesis 3: Compulsivity is linked to preemptive avoidance of bias sensitive situations.**   To test this hypothesis, we used as dependent variable the self-reported preemptive avoidance (self-report Q3). In the discovery sample, higher compulsivity was linked to more preemptive avoidance (hypothesis 3A). Finally, we tested directly the relationship between the different behaviours and self-reports affected by compulsivity. In the discovery sample, the relationship between the behavioural measure cost insensitivity during later searches (regression weight $b_{Cost}$) and self-reported preemptive avoidance (self-report Q3) was mediated by metacognitive awareness, i.e., the self-reported over chasing (self-report Q2)—hypothesis 3B.

## Power analysis

To determine the sample size required for the confirmation sample, we performed a Bayesian power analysis for each of the Bayesian analyses listed above and a non-Bayesian power calculation for the mediation analysis in hypothesis 3B (Fig 6). Sample sizes were selected to give at least 95% power for each analysis. For the Bayesian power analysis, we repeatedly simulated samples of data given the parameter estimates from our discovery sample (respecting the covariance

**Table 4. Summary of the hypotheses and tests.**

| Research question | Hypothesis | Statistical test | Outcome measure | Power | Outcome |
|---|---|---|---|---|---|
| H1 Apathy is linked to decision inertia | H1A Apathy is linked to greater decision inertia | Regression weight relating apathy to b_#PrevSearches (from the analysis "Decision model—later searches") | Significant: 95% 1-tailed Bayesian credible interval excludes zero; Not significant: 95% credible interval includes zero | 0.950 | Confirmed |
| | H1B Apathy is linked to greater response speeding with the number of searches already done | Regression weight relating apathy to b_#PrevSearches (from the analysis "RT model—later searches") | | 0.998 | Not tested |
| H2 Compulsivity is linked to a bias to oversearch due to cost insensitivity | H2A Compulsivity is linked to reduced sensitivity to costs encountered during the later searches (i.e., after the initial search) | Regression weight relating compulsivity to b_Cost (from the analysis "Decision model—later searches") | | 0.982 | Not confirmed |
| | H2B Compulsivity is linked to increased self-reported chasing | Regression weight relating compulsivity to self-reported chasing (self-report Q2) | | 0.996 | Confirmed |
| | H2C Compulsivity is linked to increased slowing with costs encountered during the later searches | Regression weight relating compulsivity to b_Cost (from the analysis "RT model—later searches") | | 0.996 | Not tested |
| H3 Compulsivity is linked to preemptive avoidance | H3A Compulsivity is linked to increased self-reported preemptive avoidance | Regression weight relating compulsivity to self-reported preemptive avoidance (self-report Q3) | | 0.996 | Confirmed |
| | H3B Relationship between cost insensitivity and preemptive avoidance is mediated by metacognitive awareness | Mediation analysis of cost insensitivity (b_Cost from "Decision model—later searches") to self-reported preemptive avoidance (self-report Q3) by self-reported chasing (self-report Q2) | Sig.: $p$-value of mediation $<0.05$ (1 sided); not significant: $p$-value $> 0.05$ | 1.000 | Confirmed |

All statistical tests consisted of a multiple linear regression, predicting the behavioural measure of interest based on the clinical factors of interest while controlling for all other clinical factors, as well as for age, gender, and education. Power is indicated for a sample size of 750 participants. See Methods—"Relating questionnaires to task performance and self-reports" and Fig 5 for further details. Elements of hypotheses that were based on an indexing error in the RT analysis of the discovery sample are greyed out for clarity and were not tested in the confirmation sample. The last column indicates which hypotheses were confirmed, partially confirmed or not confirmed in the confirmation sample.[m]

between the various psychological and demographic factors) and checked the percentage of samples where the Bayesian credible interval excluded zero (i.e., we calculated the power).

A summary of all differences between discovery and confirmation sample can be found in Table 5.

## Exploratory analysis

In addition to the preregistered analyses, we wanted to test whether our task behaviour and self-report measures, as a whole, had predictive power over the transdiagnostic dimensions and demographics. We did this by building and testing overall predictive models from the discovery sample as follows:

Step 1: In the discovery sample, we established which behavioural/post-task self-report measures showed significant links to the transdiagnostic/demographic measures based on the regression analyses described above (e.g., predicting behavioural measures based on all transdiagnostic and demographic measures). Our cutoff for including a regressor was whether the 95% Bayesian confidence interval included 0 or not.

Step 2: We then fit regression models to the discovery sample in which transdiagnostic/demographic measures were included as predicted variables and all behavioural/self-report measures identified in step 1 were included as predictors.

**Table 5. Differences between the discovery and confirmation sample.**

|  | Discovery sample | Confirmation sample |
|---|---|---|
| Sample size (after exclusions) | 449 | 756 |
| Number of questionnaires included | 13, mean: completion time: 25 minutes (AMI, AUDIT, BDI, BIS, FAS, FFMQ, IDS, OCI, PANAS, STAI, SPQ, TAS, and LSAS) | 7, completion time: 17 minutes (AMI, BDI, FAS, OCI, SPQ, TAS, and LSAS) |
| Questionnaire delivery | After behavioural task without further instruction | After behavioural task, instructions about importance for study and warning that questions will be repeated to check consistency of answers; warning given for each questionnaire if time to complete is lower than expected; 5 questions are repeated ("check question") |
| Exclusion criteria for questionnaire | (1) Not completing all questionnaires; (2) Scoring opposite extreme scores on questions with similar wording across questionnaires; (3) Responding faster than expected reading time on at least 2 questionnaires; (4) Difference of less than 0.5 points between reversed and nonreversed items on at least 2 questionnaires | (1) Not completing all questionnaires; (2) Difference of more than 1 point on at least 2 repeated questions (out of 5) |
| Number of trials in task | 200 (45 minutes to complete) | 150 (36 minutes to complete) |
| Statistics of relationship between psychological factors and task behaviour/reports of behaviour | Two-tailed | One-tailed for preregistered analyses; 2 tailed for exploratory analyses |
| Factor analysis technique to derive factor scores | Exploratory factor analysis | Application of factor weights derived from discovery sample |
| Construction of schedule | 10 schedules designed to be sensitive to presence or absence of behavioural effects on population as a whole (see [14]) | 10 schedules designed to give good parameter recovery (correlation of $r > 0.8$ between simulated and recovered parameters, within and between schedules) |

AMI, Apathy Motivation Index; AUDIT, Alcohol Use Disorders Identification Test; BDI, Beck Depression Inventory; BIS, Barratt Impulsiveness Scale; FAS, Fatigue Assessment Scale; FFMQ, Five Facets of Mindfulness Questionnaire; IDS, Inventory for Depressive Symptomatology; LSAS, Liebowitz Social Anxiety Scale; OCI, Obsessive-Compulsive Inventory; PANAS, Positive Affect and Negative Affect Schedule; SPQ, Short scales for measuring schizotypy; STAI, State-Trait Anxiety Inventory; TAS, Toronto Alexithymia Scale.

Step 3: We applied the models from step 2 to the confirmation sample. This thus generated predictions for each person in the confirmation sample of their transdiagnostic/demographic measures.

Step 4: We compared these predicted measures to the actually observed transdiagnostic/demographic measures to check the goodness of the predictions. For all continuous measures, this was done by correlating predicted and measured transdiagnostics/demographics. For the binary measure (gender), we used the negative log likelihood instead. Significance was assessed using permutation tests (10,000 permutations; randomly shuffled predictions).

In future studies, such analyses could easily be prespecified (as has been done before [70]). However, even without preregistration this approach is appealing precisely because it reduces a potentially large set of regressor-to-factor relationships to a small number of sets that are each based on the discovery sample and validated on the confirmation sample, while limiting the number of tests to one per factor of interest.

## Added question to questionnaire

Based on the discovery sample results and the resulting hypothesis 2, we added a self-report measurement directly targeting cost insensitivity (S2D Fig) in addition to overchasing. However, because it was not part of the discovery sample, we did not include it in the preregistered tests. The exact wording of the cost insensitivity question (self-report Q4) was "Did you ever

feel like you spun too many times in a round because you did not take enough into account how expensive it was to spin?," and we analysed it exactly like the other self-report measures.

## Protocol deviations

[a] We added this summary after the preregistration to improve clarity.

[b] In the preregistration and the discovery sample, this was specified as "responded faster than the time it took one author (JS) to read the questions." It was changed to the present procedure to be more objective. Unfortunately, the preregistration had not been updated to reflect this.

[c] In the preregistration processR was used, but was no longer compatible with the latest version of R. Version numbers have been added for all R packages.

[d] The number of samples was increased compared to the preregistration as more were needed to achieve convergence using the prespecified criteria.

[e] We cut the following text from the preregistration "As previously [14], we also tested (using R toolbox processR) whether the relationship between decision inertia (i.e., regression weight $b_{\#PreviousSearches}$) and preemptive avoidance (i.e., regression weight $b_{AvgProspVChange}$) was moderated by awareness of decision inertia (self-report Q1). Significance will be tested at $p < 0.05$ 1 tailed in the confirmation sample (and 2 tailed in the discovery sample)," as we did not run those analyses because they were not essential for any of our hypotheses.

[f] The number of samples was increased compared to the preregistration as more were needed to achieve convergence using the prespecified criteria.

[g] This summary was added after the preregistration to increase clarity.

[h] The level of detail specifying the confirmatory factor analysis was insufficient. We therefore removed the old short version and added this section instead. Importantly, all steps were carried out as prespecified, i.e., we did not diverge from our analysis plan.

[i] This explanation was added for clarity but the analysis was exactly the same procedure that we had already used for the discovery sample.

[j] This explanation was added after the preregistration as we discovered an indexing error in the discovery sample during the analysis of the confirmation sample.

[k] We removed the following text and analysis "And we will include the corresponding measure from the RT model of the later searches—hypothesis 1B. We predict that the higher the apathy, the more participants will speed up with having done more searches," as the discovery sample RT analysis was erroneous.

[l] As in H1, we removed the following text and analysis related to RT "Participants are aware of the cost insensitivity and they will try (but fail) to counteract it as it occurs, slowing them down. We predict therefore that compulsivity will be linked to a greater slowing of RT with increased cost (i.e., each person's regression weight for bCost from the RT analyses)—hypothesis 2C."

[m] The added this explanation to the table after the preregistration as we found the indexing error during the analysis of the confirmation sample.

[n] This figure still contains the RT based results for determining power as we used them when deciding on confirmation sample size.

## Ethical approval plan

Ethical approval has already been obtained for the discovery and the confirmation sample (see above).

## Supporting information

**S1 Text. Contains Supporting information Methods, Results, and Supporting information Tables A–I. Table A: Parameter recovery (schedule used for confirmation sample).** The table shows correlations (Pearson's r) between simulated ("ground truth") parameters (headers highlighted in grey) and the fitted ("recovered") parameters (headers with white background) for the decision models of the initial (top) and later (bottom) searches. All parameters showed very good recovery ($r > 0.88$, diagonal, highlighted in blue). The off-diagonal values show that confusion between parameters was very low (all $< 0.22$). We used 500 simulated participants for these results. Data in files S13 and S14 Data. **Table B: Parameter correlations for initial decisions** (confirmation sample). We ran a Pearson correlation of the hierarchically fit parameter estimates from our cognitive model on the initial decisions to look at their relationships. Even though simulations (S1 Table) showed very low parameter confusions, parameters from real participants showed in parts substantial correlations (Pearson's r), suggesting those correlations are a real relationship existing in participants, rather than an artifact of model fitting. invTemp (inverse temperature of the softmax equation linking value to choice probabilities), SearchBias (bias for or against searching). Prospective (prospective value of the model), Myopic (Myopic value of the model). AvgProspVChange (average change of prospective value per search in the future). Data in file S11 Data. **Table C: Parameter correlations for later searches** (confirmation sample). We ran a Pearson correlation of the hierarchically fit parameter estimates from our cognitive model on the later decisions to look at their relationships. Even though simulations (S1 Table) showed very low parameter confusions, parameters from real participants showed in parts substantial correlations (Pearson's r), suggesting those correlations are a real relationship existing in participants, rather than an artifact of model fitting. Abbreviations same as S2 Table. ProspVS1 (prospective value at first search). ProspVS1mAdapted (change of prospective value since first search). PrevSearches (number of previous searches). Total Cost (sum of total cost since first search). Data in file S9 Data. **Table D: Additional demographics (confirmation sample).** Participants completed the following questionnaires. AMI [1], OCI-R [2], BDI [3], Spielberger State Anxiety [4], Liebowitz Social anxiety [5], Short Scales for Measuring Schizotypy [6] ("unusual experiences" and "introvertive anhedonia" subscale), Fatigue scale [7], and Toronto alexithymia scale [8]. Depression cutoffs were (less than 13 "Minimal," less then 20 "Mild," less than 29 "Moderate," and higher or equal 29 "Severe"). Obsessive compulsion less than 21 "none," otherwise "clinically significant"). For social anxiety, we had less than 30 as "none," between 30 but less than 60 as "possible" and at and above 60 as "probable" social anxiety. For Alexithymia, we had less than 52 as none, between 52 but less than 61 as "possible" and at and above 62 as "present" Alexithymia. Fatigue was "none" below 22, "present" between 22 and 34, and "extreme" above 34. Data in file S5 Data. **Table E: Results of regressions linking clinical dimensions to decision and self-report measures controlling for medication status** (confirmation sample). As control regressors, we included age, gender and education (as in main manuscript). Additionally, here, psychoactive medication status is also included (coded as "Yes"/"No" as there were too few respondents of individual types of medications). Data in files S2, S6, S9, and S16 Data. **Table F: Results of regressions linking clinical dimensions to later search decisions.** Results of all parameters from model fit to the later search decisions and dimensions. We used 90% 2-sided CIs, to show the results of 1-sided (95% CI) tests for our preregistered hypotheses and to show what other links have any evidence. Data in files S9 and 16 Data. **Table G: Results of regressions linking clinical dimensions to initial search decisions.** Results of all parameters from model fit to the initial search decisions and dimensions. We used 90% 2-sided CIs, to show the results of 1-sided tests for our preregistered hypotheses and to show what other links have any

evidence. Data in files S11 and S16 Data. **Table H: Results of regressions linking clinical dimensions to self-report measures.** Results of all self-report measures fit to the dimensions. We used 90% 2-sided CI, to show the results of 1-sided tests for our preregistered hypotheses and to show what other links have any evidence. Data in files S2 and S16 Data. **Table I: Additional medication information (confirmation sample)**. All information for the types of medications excluded and included participants used (No and Yes). The most common medication were selective serotonin reuptake inhibitor. Data in file S6 Data. AMI, Apathy Motivation Index; BDI, Beck Depression Inventory; CI, confidence interval; OCI-R, Obsessive-Compulsive Inventory.
(PDF)

**S1 Fig. Effects on decisions and deliberation time during initial decisions (confirmation sample). (A)** Using a decision model, we tested which factors influenced the decision of whether to initiate a search. We found that participants were more likely to search when prospective and myopic values were high and that vice versa they were less likely to search when initial offer value or costs of searching were high. **(B)** We analysed the impact of different factors on RTs using a regression analysis. Of particular note, the larger the prospective value, the slower participants responded, potentially indicating more time taken to plan ahead. **(C)** When directly asked about their preemptive avoidance about half never reported using such strategies (self-report Q3). **(E)** Histogram of RTs across all participants, for initial searches on each trial (red) and later searches (blue). Note that for initial decisions there was a fixed delay of 3 to 6 seconds on each trial before participants could make responses, which is not included in the RT measured here. RTs for initial searches have an earlier mode than for later searches, but more slow RTs ("thicker tail" of the distribution). Error bars show Bayesian 95% credible intervals (2-tailed), significance is shown by credible intervals not including zero. Data of A in file 11, B in 12, C in 2, and D in 1. RT, reaction time.
(TIFF)

**S2 Fig. Effects on decisions and deliberation time during later decision sequence (confirmation sample). (A)** Once participants engaged with a sequence of searches, they had to decide how long to go on for. We found evidence for decision inertia ("# Prev. searches") that was independent of effects of "sunk cost" fallacy ("Total cost") or lack of updating of prospective value, i.e., their search strategy ("ProspVS1-Adapated"). Across all participants, we found sensitivity to costs ("Cost"). **(B)** RTs for later searches revealed that the more often participants had searched already within a given sequence, the faster they got ("#Prev. searches"). Moreover, as in the initial search (S1B Fig), the higher the cost, the slower they responded on these later searches in the sequence ("Cost"). **(C)** Model fits for the decision data, higher values indicate worse model fit (sum of log likelihoods across all participants and all trials, computed using cross-validation, relative to the best fitting model). The "full model"—the model used throughout the manuscript—provides the best fit to the data. The other models were derived from the "full model," leaving out individual components of the model, one at a time. **(D)** Participants were also asked about their own task behaviour, such as their perceived cost insensitivity and most reported insensitivity to the costs to some extent (question only included in the confirmation sample). **(E)** Despite the strong choice effects of decision inertia, about half of the participants reported having no such bias (self-report Q 1). **(F)** Contrary to this, about 80% of people reported having been biased towards overchasing a rewarding option at least somewhat (self-report Q2). Error bars show Bayesian 95% credible intervals (2-tailed), significance is shown by credible intervals not including zero. Data of A in file 9, B in 10, C in 15, and D to F in 2.
(TIFF)

**S3 Fig. Psychiatric symptom questionnaire results in discovery sample. (A)** Histograms of the distribution of total questionnaire scores for the OCI-R (Ai), the AMI (Aii), the BDI (Aiii), and the Liebowitz social anxiety scale (Aiv). On each histogram, cutoffs based on previously published normative data are highlighted. **(B)** Correlation matrix (Pearson's r) for all subscales included in the factor analysis. Squares highlight the factors subscales were assigned to in the factor analysis. **(C)** Loadings (i.e., the contribution of each subscale to each factor) of the factor analysis of the subscales. Highlighted in black are loadings above 0.4, for ease of visualisation. AMI, Apathy Motivation Index; BDI, Beck Depression Inventory; OCI-R, Obsessive-Compulsive Inventory.
(TIFF)

**S4 Fig. Partial correlations between clinical factors and behavioural measures (confirmation sample), controlling for other clinical, demographic and general task performance measures (see Methods—"Relating questionnaires to task performance and self-reports").** Significance was assessed using nonparametric correlations (Kendall's tau). Panel A corresponds to regression coefficients shown in Fig 4. Panels B–D correspond to regression coefficients shown in Fig 5 A–C. Note that for correlations of clinical factors with self-report questionnaires (C and D), responses needed to be treated as continuous, while in the regression analysis they were treated as ordered factors. Data of A to D in file 9, 2, and 16.
(TIFF)

**S5 Fig. Parameter recovery (confirmation sample).** We constructed 10 random schedules as detailed in the S1 Text section "Task schedule design and model validation." Scatter plots show correlations between simulated and recovered parameters for the initial search (Ai) and later searches (Ai). Different colours show the different schedules (each with $n$ = 500 simulated participants). Correlations (Pearson's r) are shown across all 10 schedules. The average correlations (Pearson's r) between regressors used in the RT regression and the decision-making models are at or below 0.7 for all variables for initial decisions (Bi) and later (Bii)." Data of A in file 13 and 14 and B in 1.
(TIFF)

**S1 Data. Raw behavioural data (choices and RT) for each participant for each trial of the decision-making task.** RT, reaction time.
(CSV)

**S2 Data. Raw questionnaire answers (for each item of each questionnaire) and post-task self-reports for each participant.**
(CSV)

**S3 Data. Summary scores for questionnaire subscales (separately for each questionnaire) for each participant.** See Fig 3B.
(CSV)

**S4 Data. Demographics for each participant.** See Table 1.
(CSV)

**S5 Data. Clinical questionnaire summed scores for each participant.** See Fig 3A and Table D in S1 Text.
(CSV)

**S6 Data. Each participant's medication status.** See Table E and Table I in S1 Text.
(CSV)

**S7 Data. Each participant's probability of searching binned by relative value.** See Fig 1E.
(CSV)

**S8 Data. Each participant's probability of searching binned by the number of previous searches and cost of searching.** See Fig 1F.
(CSV)

**S9 Data. Computational decision model parameters for each participant for data fitted on the decisions during the later searches on each trial.** See Figs 2A, 4, and 5 as well as S2A and S4–S6 Figs and Table 3.
(CSV)

**S10 Data. Computational RT model parameters for each participant for data fitted on the decisions during the later searches on each trial.** See S2B Fig and Table 3. RT, reaction time.
(CSV)

**S11 Data. Computational decision model parameters for each participant for data fitted on the decisions during the initial search on each trial.** See S1A Fig, Table 3, and Tables B and H in S1 Text.
(CSV)

**S12 Data. Computational RT model parameters for each participant for data fitted on the decisions during the initial search on each trial.** See S1B Fig and Table 3. RT, reaction time.
(CSV)

**S13 Data. Decision model simulation: simulated ("true") and recovered ("fitted") parameters for later searches of each trial.** See S5 Fig and Table A in S1 Text.
(CSV)

**S14 Data. Decision model simulation: simulated ("true") and recovered ("fitted") parameters for the initial search of each trial.** See S5 Fig and Table A in S1 Text.
(CSV)

**S15 Data. Model fits (cross-validation) for each participant for the later searches of each trial.** See S2C Fig.
(CSV)

**S16 Data. Transdiagnostic dimensions (factor scores) for each participant.** See Figs 3C, 4, and 5, S4 Fig, Tables 2 and 3 as well as Tables E–I in S1 Text.
(CSV)

## Author Contributions

**Conceptualization:** Jacqueline Scholl, Matthew F. S. Rushworth, Nils Kolling.

**Data curation:** Jacqueline Scholl.

**Formal analysis:** Jacqueline Scholl.

**Funding acquisition:** Jacqueline Scholl, Hailey A. Trier, Matthew F. S. Rushworth, Nils Kolling.

**Methodology:** Jacqueline Scholl, Hailey A. Trier.

**Project administration:** Jacqueline Scholl.

**Software:** Jacqueline Scholl, Hailey A. Trier.

**Supervision:** Matthew F. S. Rushworth, Nils Kolling.

**Validation:** Jacqueline Scholl, Hailey A. Trier.

**Visualization:** Jacqueline Scholl.

**Writing – original draft:** Jacqueline Scholl, Nils Kolling.

**Writing – review & editing:** Jacqueline Scholl, Hailey A. Trier, Matthew F. S. Rushworth, Nils Kolling.

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
