## [Editor Report · Decision Letter 0]

11 Nov 2019

Dear Dr Scholl, 

Thank you for submitting your manuscript entitled "Should I stick with it or move on? The effect of apathy and compulsivity on planning and stopping in sequential decision making" for consideration as a Registered Report by PLOS Biology.

Your manuscript has now been evaluated by the PLOS Biology editorial staff, as well as by an academic editor with relevant expertise, and I'm writing to let you know that we would like to send your submission out for external peer review.

IMPORTANT: The Academic Editor suggests that you include two additional items in the main manuscript before we sent it out to review. Please could you do this in addition to uploading the additional metadata:

AE's comments: "I would recommend that the authors make some minor revisions before in-depth review. First, the power analysis section that currently resides in the supplement should be moved into the main text. Second, I would recommend the authors include a table toward the end of the Stage 1 manuscript that shows the link between each hypothesis or sub-hypothesis, its corresponding Bayesian sampling plan (linked to that specific hypothesis and analysis), the specific analysis that will test that hypothesis, and contingent interpretation depending on different outcomes (making clear which outcomes would confirm or disconfirm the hypothesis). At present these links are not sufficiently clear for me, and would likely be an area of concern for reviewers."

ALSO IMPORTANT: Before we can send your manuscript to reviewers, we need you to complete your submission by providing the metadata that is required for full assessment. To this end, please login to Editorial Manager where you will find the paper in the 'Submissions Needing Revisions' folder on your homepage. Please click 'Revise Submission' from the Action Links and complete all additional questions in the submission questionnaire.

Please re-submit your manuscript within two working days, i.e. by Nov 13 2019 11:59PM. Do let us know if you need more time.

Kind regards,

Roli Roberts

Senior Editor

PLOS Biology

---

## [Decision Letter · Decision Letter 1]

7 Jan 2020

Dear Jacquie,

Thank you very much for submitting your manuscript "Should I stick with it or move on? The effect of apathy and compulsivity on planning and stopping in sequential decision making" for consideration as a Preregistered Research Article (Stage 1) at PLOS Biology. Your manuscript has been evaluated by the PLOS Biology editors, an Academic Editor with relevant expertise, and by three independent reviewers.

You'll see that the reviewers are broadly positive about your study, but each raises a number of requests for clarification, additional methodological detail, etc. In light of the reviews (below), we will not be able to accept the current version of the manuscript, but we would welcome re-submission of a much-revised version that takes into account the reviewers' comments. We cannot make any decision about publication until we have seen the revised manuscript and your response to the reviewers' comments. Your revised manuscript is also likely to be sent for further evaluation by the reviewers.

We expect to receive your revised manuscript within 2 months. Please note that I've also included at the foot of this email an updated version of the Guidelines, and the Stage 1 Template (which I don't think I sent you before). Many thanks for bearing with us as we iron out the wrinkles in our process for this new article type.

**IMPORTANT - SUBMITTING YOUR REVISION**

*NOTE: In your point by point response to to the reviewers, please provide the full context of each review. Do not selectively quote paragraphs or sentences to reply to. The entire set of reviewer comments should be present in full and each specific point should be responded to individually, point by point.

*Re-submission Checklist*

*Published Peer Review*

*PLOS Data Policy*

*Blot and Gel Data Policy*

Best wishes,

Roli

Senior Editor

PLOS Biology

REVIEWERS' COMMENTS:

Reviewer #1:

My understanding is that this is a "stage 1" paper that essentially provides context, study design, and pilot data. I review on that basis.

The authors plan to examine decision-making and compulsivity. The topic is very important and interesting. I enjoyed reading the paper and feel it has a number of novel aspects. 

Comments: 

1. The authors have already conducted a large study (n=449). Please clarify how the planned study is different - if it is simply collecting more data using an identical protocol, and the same recruitment pool (albeit in a new sample), then the presented data here aren't pilot data but a large part of an already substantially advanced study. This is fine but if so I don't think this is a 'stage 1' paper in the sense I have understood it (my understanding is that stage 1 is essentially "pre-registration" of methods for a study you plan to do in advance of doing the study). Put differently, if the study is already quite advanced in progress, shouldn't all the data just be reported in one paper, as it isn't "pre-registration"? 

2. "We collected questionnaires with measures covering a wide range of psychiatric symptoms....."

Only one type of compulsivity is considered in the study design (OCD) as far as I can discern. I appreciate some papers in the literature uses OCD symptom scales (e.g. OCI; Padua...) as a putative 'trans-diagnostic' measure but it isn't really, as all the items are specifically to do with OCD and not the vast range of other types of compulsive symptoms (including those mentioned by the authors). I would encourage the authors to incorporate a broader range of symptoms into their 'compulsivity' measure. Or if they focus on OCD, this is fine, but actually it should be referred to as "OCD-related compulsivity", rather than a trans-diagnostic measure. 

3. More detail should be provided about how the participants (and future participants) were recruited and selected. People doing online platforms may be unrepresentative of the general population. Also give thought to: are data collected online clinically valid? Is the cognitive task valid administered over the Internet? This may be the case but please provide more / clearer information. It would be important to list clearly as a limitation that online data collection is not the gold standard and may not generalise to findings collected in-person using trained raters, objective measures, and supervised cognitive tests. 

4. Consider whether exclusion based on questionnaire responses, as listed, might be too strict. You will probably knockout people in your sample who are particularly compulsive and/or have attentional abnormalities (common in the setting of compulsivity). Best would be to undertake analyses with full datasets and compare findings using these more rigorous criteria. That way you address the issue but don't have to worry about biasing your sample. 

5. Statistical approach is very convoluted for the decision-making task and its outcome measures. What will these measures mean for clinicians and patients? How easy will it be for other groups to replicate any results, unless they have a PhD in statistics? How many complex statistical assumptions are made that may or may not be valid? This is the reviewer's clarion call to the authors for well-validated straight forward decision-making task with clear easy-to-understand outcome measures. The reviewer appreciates he appears to be fighting a losing battle in the field of decision-making in this regard. If it really is necessary to use such a complex approach, please explain why, and provide an overview of the steps in language that a typical person reading a psychiatric or neuroscience journal would understand; aside from also providing the very detailed information too. 

6. CFA will be done in independent data-set; correct?

7. Have all confounds be considered - what about IQ, years of education, psychoactive medications, etc.?

Reviewer #2:

[identifies himself as Jiaxiang Zhang]

[The importance of the research question(s)]

The current pre-reg report aims to collect behavioural responses from a large online sample (>650). The sequential decision-making task to be used is from the authors' previous study, which ensures feasibility of the proposed study and validity of data collection. The large online sample enables addressing new important questions, specifically how apathy and compulsivity traits may relate to differences in choice behaviour across participants.

[The logic, rationale, and plausibility of the proposed hypotheses]

The ms provided a good rationale for the proposed research, including the need for using transdiagnostic trait measures. Three main hypotheses have been identified, relating (1) apathy to decision inertia, (2) compulsivity to cost insensitivity and (3) compulsivity to self-report measures.

I hope the authors could clarify the following two issues:

1. The ms referred to clinical populations (OCD and depression) when justifying the hypotheses. Can the current and proposed results rule out an alternative explanation that the trait-performance linear association is mainly/disproportionally contributed by those individuals with more extreme trait measures (as in Fig. 2A)?

2. The link between high apathy and excessive effort is indeed counterintuitive. I wonder why, in the current sequential decision task, participants with high apathy trait did not exhibit any avolitional types of behaviour, e.g., longer RT in the first decision (initial search) of each trial? 

[ The soundness and feasibility of the methodology and analysis pipeline ]

The authors conducted rigorous analysis on an existing online dataset, with detailed description of methods and power analysis for the proposed study. The protocol (paradigm, hierarchical modelling and analyses) is technically sound and appropriate for testing the stated hypotheses.

The author should provide a more explicit list on all the differences in data/analysis between the discovery (existing) and confirmation samples, as they scattered across the current ms (one-tailed/two-tailed, changes in questionnaires, changes in schedules, etc.).

[ Whether the clarity and degree of methodological detail is sufficient to exactly replicate the proposed experimental procedures and analysis pipeline.]

I am confused why "… 10 random task schedules …" were constructed. Will a subset of participant be tested with an identical task sequence, or will each participants be presented with an unique schedule? 

[ Whether the authors have pre-specified sufficient outcome-neutral tests for ensuring that the results obtained are able to test the stated hypotheses, including positive controls and quality checks]

Yes

[Minor]

P25. "Participants were did not adapt …"

Reviewer #3:

[identifies himself as Christoph W. Korn]

Scholl et al. propose an exciting study for a registered report: In a large online study, they use a validated sequential risky decision-making task in conjunction with a computational model to dissociate effects of apathy and compulsions and then link these to individual differences based on a transdiagnostic clustering of various questionnaires (pilot sample n>440; proposed sample n=650). I and a PhD student who helped me compile this review are generally in favor of seeing this study conducted. Here is a short assessment of the criteria for Stage 1 enlisted in the reviewer instructions (followed by more detailed specific points):

a) The research questions are important from a theoretical and an applied view (e.g., theoretical: how well can constructs such as apathy and compulsivity as derived from questionnaires be linked to features from computational analyses of decision-making tasks?; applied: how could this inform research on psychiatric spectrum conditions?).

b) The logic of the hypotheses is rather valid and clear. The authors could make it a bit easier for the readers to understand the modelling approach (see comments below).

c) The study is definitively sound and feasible. The current data and results are already quite substantial and the overall analyses are appropriate and sophisticated.

d) The methodological descriptions are detailed and allow for an exact replication.

e) The authors have conducted quality tests for their pilot data and one can assume that they will do the same for the main study. Some more quality checks are proposed below.

Specific points:

1) Registered report:

a. Motivation for registered report: The authors do not clarify why they want to conduct a registered report (rather than publishing their current data as they are). Of course, there are numerous advantages to a registered report. We feel that the specific reasons should be stated briefly.

b. Approach for registered analyses: Overall the methods are quite clearly described for the current data. But is unclear to us if the authors want to change anything for the new data collection and analyses (except for some changes mentioned in the supplementary file, e.g., changing the number of questionnaires). Put differently, have we learned anything from the current data that would suggest potential improvements for the new study? If not, that's fine and it would help the readers to know what exactly changed between the first and second study, even nothing changed.

2) Computational model and hypotheses:

a. Description in introduction and main text: It seems to us that the authors want to spare readers the technical details of the model in the main text. This comes at the cost of a rather blurry understanding of how the hypotheses are tested. For example, the second paragraph of the introduction lists what the model can achieve but this remains rather fanciful without reading the methods, which makes the paper's methods appear less robust than they actually are. Some more technical details would help (similar to the very helpful Table 2). In particular, the authors should state clearly in the main text how they can dissociate H1 and H2 since both are superficially linked to "searching longer for a better option." Even readers not versed in modelling could understand that H1 relates more to the number of previous searches while H2 relates to the search costs, which vary across trials. Also, it should be made clear that H3 is based on asking participants a separate (meta-cognitive) question about the task.

b. Alternative models: Do the authors test any potentially simpler or more complex models (irrespective of the link to individual differences)? It seems to us that the authors should first show that a given model component is relevant for the whole group of participants before going into questions of individual differences.

c. Split of behavior in 1st decision to search and consecutive decisions: Could the authors give a better rationale for this in the main text. Are the RTs of the 1st decision meaningful given that there was a forced (and jittered) delay?

d. "Decision" inertia: We were initially a bit confused by this term since participants with "decision inertia" actually end up making more risky decisions. They rather seem to have a "commitment" or "closure" inertia because they don't want to commit to a particular outcome. A clarification early on would help.

e. Sunk-cost effects: Related to the previous point. The relation of task behavior to sunk-cost effects could be clearer early on.

f. Report of individual effects: When reading the results I found it initially confusing that the individual effects were reported as mean parameter estimates in a model rather than as correlations. Maybe correlations would be helpful because they allow more intuitive assessments of effect sizes.

g. "Model-free" descriptive statistics and posterior predictive checks: Some more "model-free" diagnostics would be helpful. For example, what is the average number of searches per trial? Also, posterior predictive checks for the main model variables would be needed - as well as scatter plots for the main hypotheses (and ideally individual data points or violin plots).

3) Analyses of questionnaires:

a. 4 factors: Why did the authors restrict themselves to 4 factors? How much variance do these explain?

b. Influence of "social anxiety:" Do the authors have explanations for the negative effect of social anxiety on RTs (in Figure 3B & 4B)?

4) Various minor comments:

a. The introduction is very short and mostly lists the task and the hypotheses. 

b. It could be clearer why the authors use an individual differences approach. For example, another valid approach would be to test to groups of patients that differ in apathy and compulsivity (e.g., MDD v. OCD patients).

c. How long is the experiment? How do the authors make sure that the online participants don't interrupt the experiment? We're aware that Prolific includes a 'Maximum time allowed', but we assume that many people won't be aware of this and therefore might be concerned that participants take long breaks between parts of the experiment (e.g., take a long break between the behavioral task and the questionnaires)

d. The authors exclude persons with too little variance in their answers, which is completely fine. However, since so many questionnaires are available, it could be interesting to explore the questionnaire variables of these "excluded" participants. In other words: is there anything interesting/unusual about the people who had little variance?

e. In the power analyses presented in Figure 5, how can power decrease with larger sample size (e.g., 'Compulsivity x Chasing' has ~96% power at 450 participants, but only ~87% power at 550 participants)? Is this an artefact of the randomness in the simulations? If so, would more simulated runs be necessary to avoid this?

f. Although we appreciate the authors' attempt to make their research question especially relevant to current debates, it seems unlikely that sequential decision-making has become dramatically more common for animals and humans since 01.01.2000 (see 2nd sentence in introduction and their explicit emphasis on the 21st century). It should also be added that humans are animals, so that distinction seems somewhat redundant

g. The authors state multiple times that they want to get more ecological validity and that sequential decision-making is more realistic and that the dimensional approach is more realistic, but the last point is not so obvious: why exactly is the dimensional approach more realistic than just the questionnaires? Isn't the whole point of a well-designed questionnaire to learn something about someone that can help make predictions about their real-life behavior?

UPDATED GUIDELINES AND TEMPLATE:

Guidelines for Authors

Registered Reports/Pre-Registered Research Articles are a form of empirical article offered at PLOS Biology in which study rationale, methods and proposed analyses are reviewed prior to research being conducted. High quality protocols are reviewed for technical soundness of the proposed methodology, and provisionally accepted for publication before data collection commences. This format of article is designed to minimise publication and reporting bias, while also maximising study quality by focusing peer review on the importance of the research question/theory and rigour of the proposed methodology. It also allows complete flexibility to conduct exploratory (unregistered) analyses and report serendipitous findings. Pre-Registered Research Articles are offered across the full scope of empirical research at PLOS Biology.

Editorial and Peer Review Process for Pre-Registered Research Articles

Initial submissions (Stage 1 study proposals) will be triaged by the editorial team for importance of the research question using PLOS Biology’s general criteria for publication. Please see below for the template that should be followed for Stage 1 study proposals. Those that pass editorial triage will be invited for full submission (to complete additional manuscript details) and sent for in-depth peer review to further assess importance of the research question and to evaluate the technical soundness of the proposed study design and methodology (Stage 1). 

Following Stage 1 peer review, manuscripts will be rejected, offered the opportunity to revise the study proposal, or accepted in principle. Study proposals that meet our high standards of importance and scientific rigour will be issued an in-principle acceptance decision (perhaps after revision/s, as needed), indicating that the article will be published pending completion of the study. Stage 1 protocols are not published following upon an in-principle acceptance. Instead they are held and integrated into a single completed ‘Pre-Registered Research Article’ following review and acceptance of the final Stage 2 manuscript.

Following a Stage 1 in-principle acceptance decision, authors are required to register their approved protocol on the Open Science Framework (https://osf.io/) or other recognised repository, either publicly or under private embargo until submission of the Stage 2 manuscript. Accepted protocols can be quickly and easily registered using a tailored mechanism for Registered Reports on the Open Science Framework: https://osf.io/rr/ Authors then proceed to conduct the study, adhering exactly to the peer-reviewed and approved Stage 1 study design. When the study is complete the authors will submit their finalised manuscript, including Results and Discussion sections, for review (Stage 2). Editorial decisions will not be based on the perceived importance or novelty of the results obtained when completing the Stage 2 study. 

Any deviation from approved Stage 1 study procedures (after in-principle acceptance), regardless of how minor it may seem to the authors, could lead to rejection of the manuscript at Stage 2. In cases where the pre-approved protocol is altered after in-principle acceptance due to unforeseen circumstances (e.g. change of equipment or unanticipated technical error), the authors must consult the editors immediately for advice, and prior to the completion of data collection. Minor changes to the protocol may be permitted according to editorial discretion. In such cases, the deviation must be reported in the Stage 2 submission.

Submitting a Stage 1 Manuscript

Stage 1 submissions should include the manuscript (details below) and a cover letter. 

Authors are welcome to submit presubmission inquires for advice on the likely suitability of a study for PLOS Biology by emailing biology_editors@plos.org. However, please note that the editorial team cannot officially commit to sending manuscripts for in-depth review until a complete Stage 1 initial submission has been submitted online.

The cover letter should include:

A brief scientific case for consideration. 

A statement confirming that all necessary support (e.g. funding, facilities) and approvals (e.g. ethics) are in place for the proposed research. Note that manuscripts will generally be considered only for studies that are able to commence immediately; however authors with alternative plans are encouraged to contact the journal editors for advice.

An anticipated timeline for completing the study if the study proposal is approved and accepted.

A statement confirming that, following Stage 1 in principle acceptance, the authors agree to register their approved protocol on the Open Science Framework (https://osf.io/) or other recognised repository, either publicly or under private embargo until submission of the Stage 2 manuscript. Accepted protocols can be quickly and easily registered using a tailored mechanism for Registered Reports on the Open Science Framework: https://osf.io/rr/

A statement confirming that the authors agree to share their data, in accordance with the PLOS Data Availability Policy, and laboratory log, if needed, for all published results.

Stage 1 Manuscript Template (Study Proposal)

Stage 1 manuscripts should include the following sections:

Abstract

Brief note of question being tested, the relevance, and the proposed investigative approach. (Note: This section should have an additional outcome paragraph added in Stage 2 manuscripts).

Introduction

A review of the relevant literature that motivates the research question and a full description of the experimental aims and hypotheses. Please make sure to enumerate the specific hypotheses. Please note that this section cannot be altered at Stage 2 (see below) after a Stage 1 in-principle acceptance.

Materials and Methods

Please note that this section cannot be altered at Stage 2 (see below) after a Stage 1 in-principle acceptance.

Protocol details (repeat as necessary for all protocols being proposed): Sampling, Cell Lines / Organisms (i.e. experimental population), Materials & Reagents, Procedure/Intervention, Deliverables. Experimental procedures should be provided in sufficient detail to allow another researcher to repeat the methodology exactly, without requiring further information.

Sampling plan (e.g. power calculations or Bayesian sampling methods etc.) should be included unless clearly not appropriate. Please include details of criteria for data inclusion and exclusion (e.g. outlier extraction); procedures for objectively defining exclusion criteria due to technical errors or for any other reasons must be specified, including details of how and under what conditions data would be replaced. Please also detail when data collection would cease e.g. sample size, number of observations etc.

Proposed analysis pipeline, including all preprocessing steps, and a precise description of all planned analyses, including appropriate correction for multiple comparisons. Any covariates or regressors must be stated. Where analysis decisions are contingent on the outcome of prior analyses, these contingencies must be specified and adhered to. Only pre-planned analyses can be reported in the main Results section of Stage 2 manuscripts. However, unplanned exploratory analyses will be admissible in a separate section of the Results (see Stage 2 details below).

Statistics:

When relevant, studies involving Neyman-Pearson inference must include a statistical power analysis. Estimated effect sizes should be justified with reference to the existing literature. Since publication bias overinflates published estimates of effect size, power analysis must be based on the lowest available or meaningful estimate of the effect size. The a priori power must be 0.9 or higher for all proposed hypothesis tests. In the case of highly uncertain effect sizes, a variable sample size and interim data analysis will be permissible but with inspection points stated in advance, appropriate Type I error correction for ‘peeking’ employed, and a final stopping rule for data collection outlined.

Methods involving Bayesian hypothesis testing are particularly encouraged. For studies involving analyses with Bayes factors, the predictions of the theory must be specified so that a Bayes factor can be calculated. Authors should indicate what distribution will be used to represent the predictions of the theory and how its parameters will be specified. For example, will you use a uniform up to some specified maximum, or a normal/half-normal to represent a likely effect size, or a JZS/Cauchy with a specified scaling constant? For inference by Bayes factors, authors must be able to guarantee data collection until the Bayes factor is at least 10 times in favour of the experimental hypothesis over the null hypothesis (or vice versa). Authors with resource limitations are permitted to specify a maximum feasible sample size at which data collection must cease regardless of the Bayes factor, however to be eligible for advance acceptance this number must be sufficiently large that inconclusive results at this sample size would nevertheless be of major importance.

Full descriptions must be provided of any outcome-neutral criteria that must be met for successful testing of the stated hypotheses. Such quality checks might include the absence of floor or ceiling effects in data distributions, positive controls, or other quality checks. 

Any description of prospective methods or analysis plans should be written in future tense. For the Stage 2 manuscript, once the study is complete, these instances of future tense should be changed to past tense.

Summary Table

Please include a summary table that aligns each research question with the hypothesis/es used to answer the question, the sampling plan for each hypothesis (e.g. power analysis, where applicable), the specific statistical analysis/es that will be used to test the hypothesis, and a pre-specification of which outcomes will confirm or disconfirm the hypothesis (to varying degrees of strength where multiple analyses with different possible outcomes are used to interrogate one hypothesis).

Timeline 

Present anticipated timeline for completion of the study and proposed resubmission date if Stage 1 review is successful. Extensions to this deadline can be discussed with the editor.

Pilot Data

Optional. Can be included to establish proof of concept, effect size estimations, or feasibility of proposed methods and can include details of any preliminary data that have already been obtained (approach, materials and methods, results, analytical observations etc.). Any pilot studies will be published with the final version of the manuscript and will need to be clearly distinguished from data subsequently obtained at Stage 2 from the pre-approved experiments.

Secondary Analyses

Optional. If the study proposes secondary analyses of existing databases, please provide full details of the data to be analysed, its origin and any relevant information and citations with respect to the origin of the dataset and any previous analyses that have been performed. Please make clear what extent of prior observation you have had (secondary analysis of existing data may be bias-prone, which should be avoided). If this is a proposed replication study, full details of replication approach should be provided.

Data Availability Plan

Provide full details of where/how data and/or code produced will be shared (in line with the PLOS Data & Code sharing Policies).

Ethical Approval Plan

Provide details of ethical approval for animal and human subject research.

---

## [Decision Letter · Decision Letter 2]

7 Jul 2020

Dear Dr Scholl,

Thank you very much for submitting your revised Stage 1 manuscript "Should I stick with it or move on? The effect of apathy and compulsivity on planning and stopping in sequential decision making" for consideration as a Preregistered Research Article at PLOS Biology. Your revisions have been assessed by the reviewers, and I've discussed their comments with the Academic Editor.

The reviews of your manuscript are appended below. I have discussed the concern raised by reviewer #1 with the Academic Editor, who is satisfied (as are the other reviewers) that your Stage 1 Protocol meets our criteria for importance of research question and technical soundness of the study proposal. We would therefore like to invite you to complete the study, as proposed, and submit the Stage 2 manuscript. Please carefully read all the following information.

*Explanation of Decision*

This is a Stage 1 'in-principle acceptance' decision, with a commitment to publish the final Stage 2 Preregistered Research Article (after revision, if needed), pending successful completion of the study according to these Stage 1 approved methods and analytic procedures, as well as an evidence-based interpretation of the results. Please see here for review criteria for Stage 2 manuscripts:

https://journals.plos.org/plosbiology/s/reviewer-guidelines#loc-reviewing-preregistered-research-articles

Editorial decisions will not subsequently be based on the perceived importance or novelty of the results obtained during the Stage 2 study. It is critical however that you adhere exactly to this approved Stage 1 study design when performing the study. Any deviation from these experimental procedures could lead to rejection of the manuscript at Stage 2. Please consult the editors immediately for advice if you need to alter this approved study plan.

**IMPORTANT**: Please follow the link below for important information regarding the Stage 2 manuscript template and review criteria. Please carefully read the guidelines on Stage 2 data collection BEFORE performing your study and completing your Stage 2 manuscript. 

AUTHOR GUIDELINES: https://plos.io/AuthorGuidelines

*Depositing this Stage 1 Protocol*

PLOS Biology does not publish Stage 1 Protocols immediately following an in-principle acceptance. Instead they are held and integrated into a single, completed 'Preregistered Research Article' following review and acceptance of the final Stage 2 manuscript. You are however required to register this approved Stage 1 Protocol with the Center for Open Science (https://cos.io/prereg/) or another recognised repository. This may be done publicly or under private embargo until submission of the Stage 2 manuscript. Stage 1 Protocols can be quickly and easily registered using a tailored mechanism for Registered Reports (https://osf.io/rr/). Please do this now. You will need to include the URL to this deposited protocol in your Stage 2 manuscript.

*Timeline*

We understand that carrying out the study will require a significant length of time and and are willing to allow you up to nine months to perform the study (we note that you stipulate six months in your manuscript, but we assume that the ongoing pandemic may result in some delays). Please email us at 'plosbiology@plos.org' to discuss this if you have any questions or concerns, or to discuss an alternate timeline.

At this stage, your manuscript remains formally under active consideration at our journal. Please notify us by email if you do not wish to submit a Stage 2 manuscript or wish to pursue publication elsewhere, so that we may end consideration of the manuscript at PLOS Biology. 

*Resubmission Checklist*

Before submitting the Stage 2 manuscript, please review the following resubmission checklist: https://plos.io/Biology_Checklist

*Published Peer Review*

*PLOS Data Policy*

Please note that as a condition of publication, PLOS' data policy (http://journals.plos.org/plosbiology/s/data-availability) requires that you make available all data used to draw the conclusions arrived at in your manuscript. Please note that for this article type, the raw data itself should be archived and made freely available in a public repository rather than submitted as supplementary material. Please make sure to read the Stage 2 submission guidelines online regarding how this data should be annotated and appropriately time stamped to show that data was collected after this Stage 1 in-principle acceptance and not before.

*Blot and Gel Data Policy*

To enhance the reproducibility of your results, we recommend that, if applicable, you deposit your laboratory protocols in protocols.io, where a protocol can be assigned its own identifier (DOI) such that it can be cited independently in the future. For instructions see: https://journals.plos.org/plosbiology/s/submission-guidelines#loc-materials-and-methods

Thank you again for your submission to PLOS Biology. We hope that our editorial process has been constructive thus far, and we welcome your feedback at any time. Please don't hesitate to contact us if you have any questions or comments.

Sincerely,

Roli Roberts

Senior Editor

PLOS Biology

REVIEWERS' COMMENTS:

Reviewer #1:

My only remaining point is (also raised in my original review): is this really a pre-registration? 

The rationale for a pilot in several hundred people in order to do a power calculation to collect more data is not convincing - it is pretty obvious that a study in 700 or so people would have sufficient power to test specific hypotheses - e.g. based on power calculations not requiring data (e.g. assuming a medium or small effect size). The idea of collecting two samples to cross-validate results is of course excellent and to be applauded. But isn't this really a single study that should be submitted as a single full data paper? Pre-registration is done before a research study, not after it. As such, publishing this as a pre-registration paper seems inappropriate/inaccurate to me. But I leave that judgement to the editor. 

The authors have addressed the other issues raised in my original review. For this reason, my recommendation is accept, but subject to the editor confirming that this is a pre-registration paper; and if not, I recommend it be published as a standard paper. 

For future work, I would encourage the authors to think beyond only OCD in terms of compulsivity. Are the factor scores provided by e.g. Padua/OCI (e.g. washing/checking, contamination fears, ordering, thoughts of harm to self/others) really telling us about trans-diagnostic compulsivity? Many of these aren't relevant to any other compulsive disorder apart from OCD. 

Reviewer #2:

[identifies himself as Jiaxiang Zhang]

The authors addressed all my concerns in this revision. I look forward to the results of this pre-reg study.

Reviewer #3:

[identifies himself as Christoph W. Korn]

We would like to thank the authors for addressing all of our comments. For example, Figure R5 is very helpful. The responses regarding alternative models and the transdiagnostic approach are convincing and very well taken.

We wish the authors all the best for the data collection and look forward to seeing the results.

---

## [Editor Report · Decision Letter 3]

21 Jul 2021

Dear Jacquie,

Many thanks for submitting the Stage 2 version of your manuscript "Should I stick with it or move on? The effect of apathy and compulsivity on planning and stopping in sequential decision making" for consideration as a Preregistered Research Article at PLOS Biology. 

IMPORTANT: In order to make the path as smooth as possible, the Academic Editor has asked me to get you to do the following things before we send it back out for Stage 2 review:

1. Please include the OSF URL for the registered Stage 1 MS, instead of your current placeholder. If the OSF deposition is under embargo, you now need to make it public, especially as the reviewers may need to cross-check the current version against it. (see https://plos-marketing.s3.amazonaws.com/Marketing/Biology+Preregistered+Articles+Guidelines+for+Authors.pdf). I will also include this link in the reviewer invitations.

2. Note that I will then check that the OSF-registered version only differs from our Stage 1 accepted version with regard to the approved change (a tweak to exclusion criteria) that you made after acceptance.

3. Please provide a new "Track Changes" (mark-up version) that also includes tracking of all text changes in the Abstract, Introduction and Method sections (at the moment you've done so for the Methods section, but there are also changes to the Abstract that aren't tracked; I haven't checked the Introduction). Ideally you should mark up the actual changes, not just highlight the changed paragraphs. Again, I will re-check this before review.

4. The Academic Editor suggests that you update Table 4 (the design table) to include an extra column to the right called "Outcome" which simply reports whether the hypothesis was confirmed or not confirmed. The reviewers and readers will probably find this helpful.

We expect to receive your revised manuscript within 1 week.

**IMPORTANT - SUBMITTING YOUR REVISION**

In addition to a clean copy of the manuscript, please also upload a 'track-changes' version of your manuscript that specifies the edits made. This should be uploaded as a "Related" file type. 

*Resubmission Checklist*

*Published Peer Review*

*PLOS Data Policy*

*Blot and Gel Data Policy*

Best wishes,

Roli

Roland Roberts

Senior Editor

PLOS Biology

rroberts@plos.org

---

## [Decision Letter · Decision Letter 4]

25 Oct 2021

Dear Jacquie,

Thank you for submitting your revised Preregistered Research Article entitled "Should I stick with it or move on? The effect of apathy and compulsivity on planning and stopping in sequential decision making" for publication in PLOS Biology. I have now obtained advice from two of the original reviewers and have discussed their comments with the Academic Editor. 

Based on the reviews, we will probably accept this manuscript for publication, provided you satisfactorily address the remaining points raised by the reviewers. Please also make sure to address the following data and other policy-related requests.

IMPORTANT: Please attend to the following:

a) Please attend to the reviewers' remaining requests. Note that the Academic Editor asked me to transmit the following important caveat: "Concerning the comment from rev #2: 'Although exploratory analyses are clearly marked as they are, I wonder if the authors could unpack the results in Intro/Discussion a bit more, as they can be as useful as confirmatory results for future studies.' This is a useful suggestion but it's very important to make clear to the authors (in the decision letter) to interpret this remark solely in terms of the Discussion, NOT the Intro. We don't want the authors making any further changes to the Introduction at this point unless it's to correct factual errors or typos. So this kind of contextualisation and interpretation should be reserved for the Discussion."

b) Please could you remove the first sentence of your title to make it more succinct and direct: "The effect of apathy and compulsivity on planning and stopping in sequential decision making"

c) Please address my Data Policy requests below; specifically, please supply numerical values underlying Figs 1DEF, 2ABCD, 3ABC, 4, 5ABC, 6, S1ABCD, S2ABCDEF, S3ABC, S4ABCD, S5, and cite the location of the data clearly in each relevant Fig legend ("The data underlying this Figure may be found at https://..."). You currently say that these data will be made available on OSF upon acceptance, but this is not satisfactory; we cannot editorially accept your paper until we have checked data availability. You will also note that both reviewers raise this point.

We expect to receive your revised manuscript within two weeks. 

*Published Peer Review History*

*Early Version*

Best wishes,

Roli

Senior Editor,

rroberts@plos.org,

PLOS Biology

DATA POLICY:

Regardless of the method selected, please ensure that you provide the individual numerical values that underlie the summary data displayed in the following figure panels as they are essential for readers to assess your analysis and to reproduce it: 1DEF, 2ABCD, 3ABC, 4, 5ABC, 6, S1ABCD, S2ABCDEF, S3ABC, S4ABCD, S5. NOTE: the numerical data provided should include all replicates AND the way in which the plotted mean and errors were derived (it should not present only the mean/average values).

DATA NOT SHOWN?

REVIEWERS' COMMENTS:

Reviewer #2:

[identifies himself as Jiaxiang Zhang]

 I have reviewed the Stage 1 report of this study. It is great to see the confirmatory results in the current report. In this manuscript, the intro, rationale, and hypotheses are the same as the stage 1 protocol, making it a high-quality stage 2 submission. The authors did a great job in adhering to original experimental procedures and analyses plans. In the case of potential inconsistency (e.g., factor analyses), the authors provided convincing evidence to stick with the original analysis plan. Overall, beyond the obvious interest to the decision-making/cognitive psychology community, I believe this study will set a nice example on how to combine a novel experimental paradigm with rigorous pre-registered analyses. Below are just a few minor comments.

1. Table 3. Although exploratory analyses are clearly marked as they are, I wonder if the authors could unpack the results in Intro/Discussion a bit more, as they can be as useful as confirmatory results for future studies. What do they mean, what can we learn from them, or how can they be extended in future studies? For example, cost is not a reliable predictor (which I think is the same predictor in hypothesis 2A?) of compulsivity, but "chasing" is. There are however other task-based measures that, together, yield a better predictive model for compulsivity. Does this mean trait measure of compulsivity is associated with multifaceted behaviour and meta-cognitive processes, instead of specific correlations highlighted by the main hypotheses? Similarly, could the authors comment on those task-based regressors predicting age/gender? Are they in line with the current literature?

2. Page 22. "95% % CI_lower:1.081, hypothesis 2C …" Do the authors mean hypothesis 2B here? As 2C refers to the hypothesis on RT.

3. Table 4. I suggest having a clear outcome report for each sub hypothesis. I.e., H1B/H2C is n/a nor not tested. H2A is not supported but H2B is confirmed (avoiding the ambiguous "partially confirmed"). 

4. Page 44 "Step 3: We applied the models from step to the confirmation sample"  "from step 2" ?

[re PLOS' data availability policy]

The authors claimed that the data will be made available after acceptance. As a result, data availability/time stamps cannot be reviewed at this stage.

Reviewer #3:

[identifies himself as Christoph W. Korn]

In my view, Scholl and colleagues have provided a very comprehensive and detailed Stage 2 manuscript. Given this complexity, it took me longer than expected to carefully assess the criteria listed for Stage 2 manuscripts. I did not find any flaws. The few deviations from Stage 1 are explained (e.g., the RT analyses). In my view, only the last criterion is not yet fulfilled since the data is not yet freely accessible. 

Below, I list a few minor suggestions for improving the clarity of the manuscript.

1. Figure 1F: Why does the model fit get worse with increasing numbers of previous searches? Is this simply because there are less data points available? Is there any particular reason for the "peak" at "6 previous searches" for "cost 15?"

2. In some places, the distinction between self-reported "pre-emptive avoidance" and the parameter 'AvgProspVChange' is a bit confusing.

a. At first, it sounds as if the authors just measured a self-report but it later becomes clear that they also have decision variable.

b. It was unclear to me why the authors did not look into 'AvgProspVChange' (see footnote e). This could be an interesting exploratory analysis.

c. In the abstract, the authors write: "In addition, this awareness allowed them to report pre-emptively avoiding situations related to the bias." The question that comes to mind immediately is: "Did participants actually pre-emptively avoid those situations?"

3. A few details could be mentioned briefly or explained more clearly in the results (and not "just" in the methods).

a. How high are the correlations between decision variables? (below .7).

b. The distinction in the analyses between the first and the later searches could be stressed a bit more in the results.

c. It could simply be stated that all results from the discovery sample are reported in the Supplemental Material and links to the corresponding figures/tables could be given.

4. Style:

a. Figure 3B (and others): The labels are difficult to read. Usually, the columns are ordered in the same way as the rows and therefore the main diagonal runs from upper left to lower right. Maybe, the easiest way is to just show the lower (or upper) triangular part of the matrix.

b. In the abstract (and also later), the term "perceived insensitivity" sounds ambiguous. "Self-reported" might be a better word.

c. In Table 4, "partially confirmed" could be replaced by a statement specifying the difference between the discovery and the confirmation samples.

d. Figure 1: The figure is quite complex and the caption is very long. Splitting this up could help.

e. There are some typos or omissions: e.g., "Batchelor," "large set regressor-to-factor relationships"

f. The title for the supplements is different.

---

## [Editor Report · Decision Letter 5]

3 Feb 2022

Dear Jacquie,

On behalf of my colleagues and the Academic Editor, Christopher Chambers, I'm pleased to say that we can in principle accept your Preregistered Research Article (our very first!) "The effect of apathy and compulsivity on planning and stopping in sequential decision making" for publication in PLOS Biology, provided you address any remaining formatting and reporting issues. These will be detailed in an email that will follow this letter and that you will usually receive within 2-3 business days, during which time no action is required from you. Please note that we will not be able to formally accept your manuscript and schedule it for publication until you have any requested changes.

PRESS: We frequently collaborate with press offices. If your institution or institutions have a press office, please notify them about your upcoming paper at this point, to enable them to help maximise its impact. If the press office is planning to promote your findings, we would be grateful if they could coordinate with biologypress@plos.org. If you have not yet opted out of the early version process, we ask that you notify us immediately of any press plans so that we may do so on your behalf.

Sincerely, 

Roli

Roland G Roberts, PhD 

Senior Editor 

PLOS Biology

rroberts@plos.org